Resource

# Clonal landscape of autoantibody-secreting plasmablasts in COVID-19 patients

Shuhei Sakakibara[1],* , Yu-Chen Liu[2],* , Masakazu Ishikawa[2,3], Ryuya Edahiro[4,5,6], Yuya Shirai[4,5,6], Soichiro Haruna[1], Marwa Ali El Hussien[1], Zichang Xu[7], Songling Li[7], Yuta Yamaguchi[4,8] , Teruaki Murakami[4,8], Takayoshi Morita[4,8], Yasuhiro Kato[4,8] , Haruhiko Hirata[4], Yoshito Takeda[4], Fuminori Sugihara[9], Yoko Naito[10] , Daisuke Motooka[2,10,11], Chao-Yuan Tsai[1] , Chikako Ono[3,12,15], Yoshiharu Matsuura[3,12,15], James B Wing[3,13,15] , Hisatake Matsumoto[3,14], Hiroshi Ogura[3,14], Masato Okada[15], Atsushi Kumanogoh[3,4,8,11,15,16] , Yukinari Okada[3,5,6,11,15,17,18], Daron M Standley[3,7,15] , Hitoshi Kikutani[1], Daisuke Okuzaki[2,3,10,11,16]

Whereas severe COVID-19 is often associated with elevated autoantibody titers, the underlying mechanism behind their generation has remained unclear. Here we report clonal composition and diversity of autoantibodies in humoral response to SARS-CoV-2. Immunoglobulin repertoire analysis and characterization of plasmablast-derived monoclonal antibodies uncovered clonal expansion of plasmablasts producing cardiolipin (CL)-reactive autoantibodies. Half of the expanded CL-reactive clones exhibited strong binding to SARS-CoV-2 antigens. One such clone, CoV1804, was reactive to both CL and viral nucleocapsid (N), and further showed anti-nucleolar activity in human cells. Notably, antibodies sharing genetic features with CoV1804 were identified in COVID-19 patient-derived immunoglobulins, thereby constituting a novel public antibody. These public autoantibodies had numerous mutations that unambiguously enhanced anti-N reactivity, when causing fluctuations in anti-CL reactivity along with the acquisition of additional self-reactivities, such as anti-nucleolar activity, in the progeny. Thus, potentially CL-reactive precursors may have developed multiple self-reactivities through clonal selection, expansion, and somatic hypermutation driven by viral antigens. Our results revealed the nature of autoantibody production during COVID-19 and provided novel insights into the origin of virus-induced autoantibodies.

## Introduction

Coronavirus disease 2019 (COVID-19) is caused by severe acute respiratory syndrome coronavirus 2 (SARS-CoV-2), which initially targets the nasal epithelium and spreads to the upper respiratory tract, lung, and other organs (Zhou et al, 2020; Khan et al, 2021). Symptoms of COVID-19 vary from mild and common cold-like to severe acute respiratory distress syndrome and multiple organ failure (Chen et al, 2020). The magnitude of immune responses against SARS-CoV-2 has been shown to be associated with COVID-19 disease severity (Chen et al, 2020; Huang et al, 2020; Lucas et al, 2020). Monocyte-derived macrophages, alveolar macrophages, DCs, and NK cells increased and were aberrantly activated in the lungs of critical COVID-19 patients (Lamers & Haagmans, 2022). CD16 expression in activated T cells in severe COVID-19 patients enabled the induction of complement-mediated, T-cell receptor-independent degranulation and cytotoxicity, causing pulmonary endothelial cell injury (Georg et al, 2022). Therefore, SARS-CoV-2 infection evokes non-specific immune responses, which can trigger severe symptoms in COVID-19 patients.

[1]Laboratory of Immune Regulation, Immunology Frontier Research Center, Osaka University, Osaka, Japan   [2]Laboratory of Human Immunology (Single Cell Genomics), Immunology Frontier Research Center, Osaka University, Osaka, Japan   [3]Center for Infectious Disease Education and Research, Osaka University, Osaka, Japan   [4]Department of Respiratory Medicine and Clinical Immunology, Osaka University Graduate School of Medicine, Osaka, Japan   [5]Department of Statistical Genetics, Osaka University Graduate School of Medicine, Osaka, Japan   [6]Laboratory of Statistical Immunology, Immunology Frontier Research Center, Osaka University, Osaka, Japan   [7]Laboratory of Systems Immunology, Immunology Frontier Research Center, Osaka University, Osaka, Japan   [8]Department of Immunopathology, Immunology Frontier Research Center, Osaka University, Osaka, Japan   [9]Core Instrumentation Facility, Immunology Frontier Research Center and Research Institute for Microbial Diseases, Osaka University, Osaka, Japan   [10]Genome Information Research Center, Research Institute for Microbial Diseases, Osaka University, Osaka, Japan   [11]Integrated Frontier Research for Medical Science Division, Institute for Open and Transdisciplinary Research Initiatives, Osaka University, Osaka, Japan   [12]Laboratory of Virus Control, Research Institute for Microbial Diseases, Osaka University, Osaka, Japan   [13]Laboratory of Human Single Cell Immunology, Immunology Frontier Research Center, Osaka University, Osaka, Japan   [14]Department of Traumatology and Acute Critical Medicine, Osaka University Graduate School of Medicine, Osaka, Japan   [15]Center for Advanced Modalities and DDS, Osaka University, Osaka, Japan   [16]Japan Agency for Medical Research and Development – Core Research for Evolutional Science and Technology (AMED–CREST), Osaka University, Osaka, Japan   [17]Department of Genome Informatics, Graduate School of Medicine, the University of Tokyo, Tokyo, Japan   [18]Laboratory for Systems Genetics, RIKEN Center for Integrative Medical Sciences, Wakō, japan

Correspondence: sakakibara@ifrec.osaka-u.ac.jp; kikutani@biken.osaka-u.ac.jp
Shuhei Sakakibara's present address is Graduate School of Medical Safety Management, Jikei University of Health Care Sciences, Osaka, Japan
*Shuhei Sakakibara and Yu-Chen Liu contributed equally to this work

Beyond the above, several studies have reported that different types of autoantibodies are produced in COVID-19 patients and are associated with disease severity and symptoms (Zuo et al, 2020; Chang et al, 2021; Tung et al, 2021; Wang et al, 2021; Wong et al, 2021; Zuo et al, 2021). Autoantibodies specific to immunomodulatory proteins, such as cytokines and chemokines, from COVID-19 patients have been shown to perturb immune response to SARS-CoV-2 (Wang et al, 2021). Several studies have suggested that antibodies to phospholipids including cardiolipin (CL) are elevated in severe COVID-19 patients and play a pathogenic role in COVID-19 coagulopathy (Zuo et al, 2020; Tung et al, 2021; Zuo et al, 2021). It has long been known that some viral infections can trigger autoantibody production (Root-Bernstein & Fairweather, 2014; Puel et al, 2022). Primary Epstein-Barr virus infection causes acute infectious mononucleosis, which is often associated with autoantibody production (Vaughan, 1995). Dengue virus infection elicits serum antibodies against not only viral proteins but also platelets and endothelial cells (Lin et al, 2001). In mice infected with certain viruses such as lymphocytic choriomeningitis virus and murine gammaherpesvirus 68, antibody responses are accompanied by autoantibody production in sera (Sangster et al, 2000; Hunziker et al, 2003; Sakakibara et al, 2020). Although various mechanisms, such as bystander polyclonal B-cell activation and molecular mimicry have long been proposed for autoantibody production during viral infections, the generation process of autoantibodies during such viral infections remains elusive.

Several studies have characterized antigen-specific mAbs derived from COVID-19 patients (Brouwer et al, 2020; Cao et al, 2020; Ju et al, 2020; Robbiani et al, 2020; Shi et al, 2020; Zost et al, 2020). Nevertheless, most of these studies focused primarily on clones derived from memory B cells, which do not necessarily reflect the serum antibody responses of patients. In this study, we analyze plasmablast-derived mAbs at the clonal level to understand the nature of antibody response in severe COVID-19, particularly the evolutionary process of virus-reactive and self-reactive clones.

## Results

### Repertoire and reactivity of plasmablast-derived mAbs in patients with COVID-19

Previous single-cell RNA sequencing (scRNAseq)-based studies by us and others highlighted robust development and clonal expansion of PBs in patients with severe COVID-19 (Bernardes et al, 2020; Schultheiss et al, 2020; Turner et al, 2021; Edahiro et al, 2023). To dissect antibody responses of COVID-19 patients at the clonal level, we performed cloning, expression, and reactivity assays of mAbs using PB-derived Ig sequences from randomly selected COVID-19 ICU patients (N = 12; mean age, 65.5 yr [min.-max., 39–80], detailed in Table S1) in the Osaka scRNAseq cohort (Edahiro et al, 2023) (Fig S1A–F). Blood was collected at the ICU admission, which was 10–15 d after symptom onset for these patients. For comparison, we investigated the reactivity of plasmablasts from healthy donors who received BNT162b2 mRNA vaccine a week before blood collection (N = 3; mean age, 28.3 yr [min.-max., 24–36]) (Table S1 and

Fig S2A). We prioritized large clones in the respective subjects for the mAb cloning to recapitulate serum antibody reactivity. In the case of subjects lacking dominant clones, several singletons were included (Table S2). In total, we generated 123 mAbs (101 and 22 from expanded clonotypes and singletons, respectively) from COVID-19 ICU patients and 66 mAbs from vaccinated donors (18 and 48 from expanded clonotypes and singletons, respectively). Among these, IgG1 and IgA1 were the major Ig classes, accounting for 56% and 28% of the clonotypes in COVID-19–derived antibodies and 42% and 26% of those in vaccine-induced antibodies (Fig S2B). The overall Ig isotype composition did not differ between expanded clonotypes and singletons (Fig S2B). Reactivity tests showed that 22% (27/123) of patient-derived mAbs reacted with SARS-CoV-2 antigens (spike [S], nucleocapsid [N], and open reading frame 8 [ORF8]) (Fig 1A and Table S2). Similarly, 23% (15/66) of vaccinated donor-derived mAbs were found to be SARS-CoV-2 S-reactive (Fig 1A and Table S2). In the assay, mAbs that did not react to these viral antigens or were sticky on BSA-coated plastic plates were considered of unknown specificity.

Half of the SARS-CoV-2 antigen-reactive clones from COVID-19 ICU patients were S-reactive. The patient-derived anti-S clones, as represented by CoV324, CoV2204, CoV2205, and CoV3179, mainly targeted the S2 region of SARS-CoV-2 S and were also cross-reactive to the S protein of seasonal coronavirus HCoV OC43 but not HCoV 229E or HCoV NL63 (Fig 1B and C and Table S2). The rest of the S-reactive clones, such as CoV901, CoV904, CoV2024, and CoV3004, reacted to SARS-CoV-2 S but not OC43 S, through targeting the RBD, S1, or a conformational epitope on the prefusion trimeric form (Fig 1B and C). Of the infection-induced mAbs, cross-reactive clones had significantly more somatic mutations than SARS-CoV-2 S-specific clones (25.0 and 9.0 VH mutations, on average, respectively) (Fig 1D). Although cross-reactive clones were entirely restricted to the IgG1 subclass, the isotypes of SARS-CoV-2 S-specific clones were diverse; that is, they were not restricted to IgG1; they also included IgM and IgA1 (Fig 1B).

In addition to anti-S antibodies, we isolated 7 mAbs reactive to the N protein, all of which targeted the N terminal domain from patient-derived clones (Fig S2C and D). Of these, CoV216 and CoV602 showed cross-reactivity to the N proteins of HCoV 299E and NL63 (Fig S2E). The rest of the anti-N clones did not exhibit reactivity to seasonal coronavirus N proteins (Fig S2E). We identified five anti-ORF8 clones that were distributed to various isotypes and had high numbers of somatic mutations (mean 21.2) (Figs 1D and S2F).

Antiviral antibodies had different degrees of somatic mutation and distribution of Ig isotypes depending on their targets and specificity. Regarding the anti-S antibodies, SARS-CoV-2–specific clones had diverse isotypes and low mutation, suggesting that they were derived directly from naïve B cells. In contrast, cross-reactive clones were all IgG1 antibodies and had more mutations than SARS-CoV-2 S-specific clones (Fig 1B and D), likely originating from pre-existing memory B cells against seasonal coronaviruses. On the other hand, despite their high mutation rates, antibodies to ORF8, a viral protein unique to SARS-CoV-2, appeared to be derived from the primary response after SARS-CoV-2 infection (Figs 1D and S2F). This was also the case for non-cross-reactive anti-N clones, which had a similar level of mutation to anti-ORF8 clones and cross-reactive anti-N clones (Figs 1D and S2E). Taken together, non-cross-

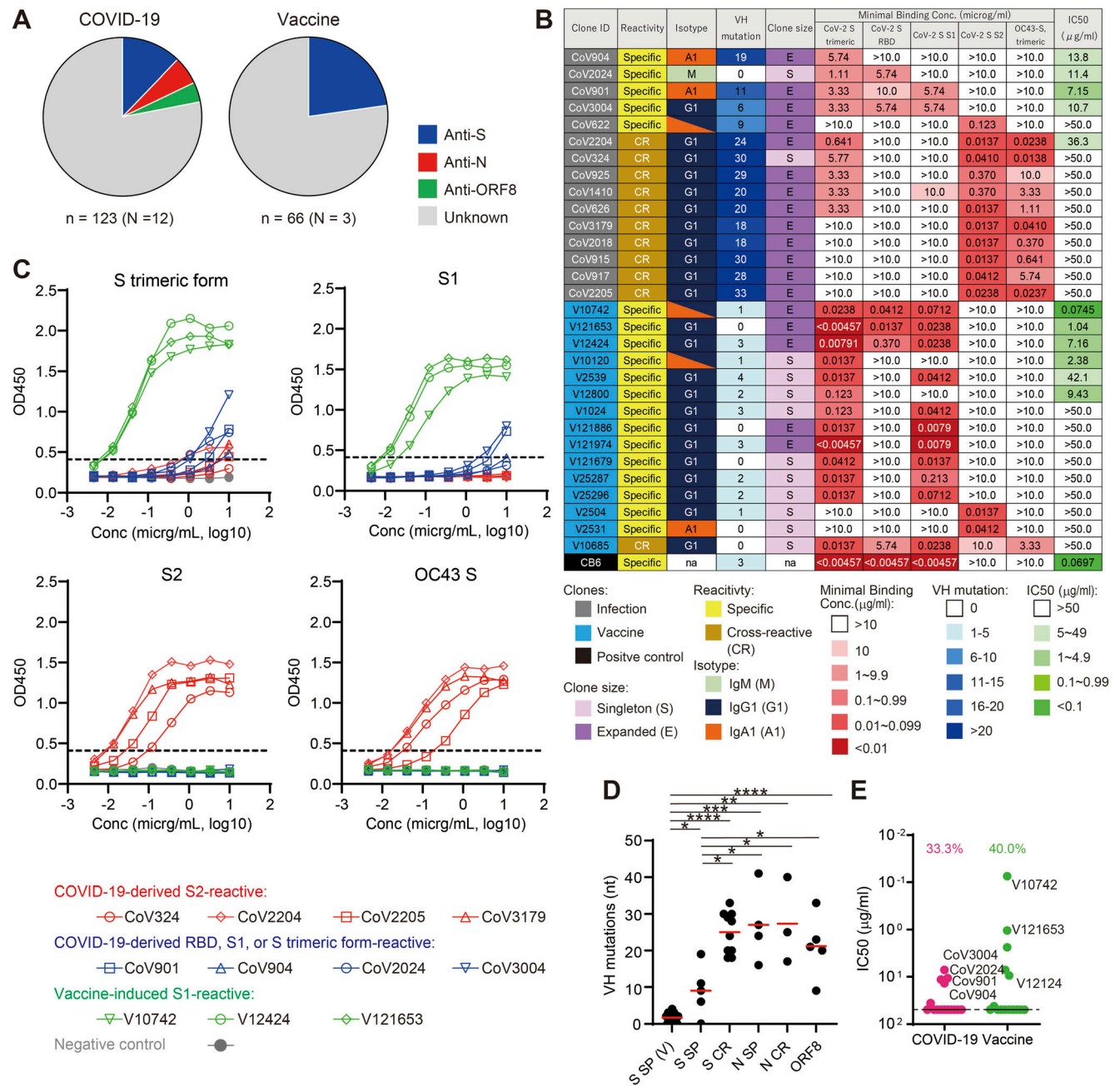

**Figure 1. Reactivity of mAbs derived from COVID-19 patients and vaccinated volunteers.**
**(A)** Proportions of antiviral clones in PB-derived mAbs. COVID-19 patients: n = 123 from 12 donors; individuals vaccinated with mRNA vaccine: n = 66 from three donors. mAb clones that did not react to viral antigens tested and were sticky on BSA-coated plastic plates were considered of unknown specificity. **(B)** Summary of anti-S clones. Minimal binding concentration was determined by ELISA. Neutralizing activity (the half inhibitory concentration [IC50]) was determined by pseudovirus neutralizing assay. **(C)** CR, cross-reactive; IgM, M; IgG1, G1; IgA1, A1; Singleton, S; and Expanded, E (C) Binding activity of representative clones to S proteins in ELISA. The dotted line indicates the threshold of positivity in each assay. **(D)** VH nucleotide mutations of vaccine-induced anti-SARS-CoV-2 S-specific (S SP[V]), infection-induced SARS-CoV-2 S-specific, infection-induced cross-reactive anti-S (S CR), SARS-CoV-2 N-specific (N SP), SARS-CoV-2 N cross-reactive (N CR) and anti-ORF8 clones. Mann–Whitney test: *≤0.05, **≤0.01, ***≤0.001, ****≤0.0001. **(E)** Neutralization activity (IC50) of mAbs against SARS-CoV-2 S pseudotyped VSVΔG is shown. **(A, B, C, E)** Clones with IC50 higher than 50 μg/ml were plotted at the limit of detection. Representative results from at least two independent assays are shown (A, B, C, E). Source data are available for this figure.

reactive anti-SARS-CoV-2 antibodies appear to have originated from primary responses which resulted in diverse Ig sequences, from near germline to highly mutated sequences comparable to those of memory-derived antibodies.

From these results, the antibody response during severe COVID-19 can be described as a mixture of primary and recall responses. This was in contrast to the response elicited by the mRNA vaccine (BNT162b). Most of the vaccine-induced anti-S clones such as

V10742, V12424, and V121653 from our donors showed no cross-reactivity with OC43 S and exclusively targeted the S1 region (Fig 1B and C and Table S2). Moreover, these mAbs were encoded by Ig genes that were nearly identical to the germline (Fig 1D). Although one vaccine-induced clone, V10742, had neutralizing activity equivalent to a potent neutralizing clone, CB6 (Shi et al, 2020), the frequency and potency of neutralizing clones were overall comparable between infection- and vaccine-induced anti-S antibodies (33% and 40%, respectively) (Fig 1E).

To verify whether the above results of mAb reactivity tests reflect serum antibodies of patients with COVID-19, we examined plasma antibody reactivity of a different COVID-19 ICU cohort obtained from the same hospital (Table S3 and Fig S3A). Anti-S IgG titers were yet to be elevated in a large fraction of patients on the day of ICU admission (T1) (Fig S3B). Anti-S titers in patients increased significantly after treatment (T2), although these antibody titers were somewhat lower than those of healthy donors following vaccination (Fig S3B). Meanwhile, anti-HCoV OC43 S IgG and anti-SARS-CoV-2 N IgG were readily detected in the plasma samples of patients at T1 but not those after vaccination (Fig S3C and D). Neutralizing activity was detectable, but weak in COVID-19 patients at T1 and later increased to a similar level to that of vaccinated donors (Fig S3E). Thus, the reactivity and function of PB-derived mAbs recapitulated those of plasma antibodies from COVID-19 patients.

## PBs producing antibodies reactive to both viral and self-antigens were robustly induced through clonal expansion during SARS-CoV-2 infection

CL antibody tests generally aid in the evaluation of systemic lupus erythematosus (SLE) and assess the risk of complications; high anti-CL antibody titers are associated with increased risk of thrombosis (Tarr et al, 2007). Several studies have shown that autoantibodies to CL and other phospholipids are elevated in severe COVID-19 patients, suggesting their potential contribution to the development of COVID-19–associated coagulopathy during the early pandemic (Zuo et al, 2020; Tung et al, 2021; Zuo et al, 2021). In agreement with previous reports, antibody titers against CL in the plasma of COVID-19 ICU patients were significantly higher than those of healthy donors without vaccination but somewhat lower than those of SLE patients (Fig 2A and Table S4). No increase in anti-CL antibody titers was observed after the second dose of BNT162b2 mRNA vaccination of the same donors (Fig 2A). When stratified by sex, anti-CL antibody titers were particularly elevated in male COVID-19 patients at T1 (Fig S4).

These results prompted us to examine a panel of PB-derived mAbs for reactivity against CL. We found that 16% (20/123) of COVID-19 ICU patient-derived mAbs were CL-reactive (Table 1 and Fig 2B). Some COVID-19–derived clones showed extremely high CL reactivity, equivalent to that of a previously isolated SLE-derived clone, 71G1 (Sakakibara et al, 2017) (Fig 2B). On the other hand, weak to moderate CL reactivity was shown in 6.0% (8/66) of vaccine-induced clones (Fig 2B). Anti-CL antibodies are one example of anti-phospholipid antibodies, which constitute a heterogeneous group of antibodies against different phospholipids. Indeed, the COVID-19–derived anti-CL clones similarly reacted to phosphatidic acid (PA) and were less reactive to phosphatidylcholine (PC), or

phosphatidylethanolamine (PE) (Fig S5). Intriguingly, 60% (12/20) of anti-CL mAbs reacted to SARS-CoV-2 antigens; these comprised six anti-S clones, five anti-N clones, and one anti-ORF8 clone (Fig 2C). In addition to anti-CL reactivity, CoV1804 and CoV1838 exhibited a characteristic pattern of staining at the nucleoli of HEp-2 cells that resembled plasma or serum anti-nucleolar antibody (ANoA) activity of some patients in our cohort (Fig 2D and E) and previous studies (Chang et al, 2021; Muratori et al, 2021). Neither anti-nuclear nor anti-nucleolar clones were found in vaccinated donor-derived mAbs, which was consistent with the absence of such reactivity in the plasma samples from vaccinated donors (data not shown).

We next assessed the clonal sizes and the frequencies of somatic mutations of autoantibody-producing cells in individual donors. Initially classified as different clonotypes within the same donor-derived clones, CoV1827 and CoV1838; CoV1901 and CoV1919; and CoV1907 and CoV1912 were determined to belong to the same clonal lineage because of their highly shared genetic features: heavy and light variable segment usage and similar HCDR3 sequences with identical amino acid length (Table 1, and Fig S6A–C). In donor SG CO OK 00018, viral antigen-reactive, CL-reactive clones, CoV1804 and CoV1827/1838 Ig-expressing cells were the fourth and 10th largest lineages, accounting for 7.3% and 1.2% of PBs, respectively (Fig 2F). Other clones in this reactivity category, CoV216, CoV324, CoV327, CoV2018, and CoV2205 also belonged to PB lineages of relatively large sizes (1–5%) and were within the top 5 largest clonotypes in their respective patients (Fig 2F). CL-reactive clonotypes, CoV330, CoV1403, CoV1901/1919, CoV1907/1912, and CoV2207 had strong anti-CL reactivity but lacked antiviral reactivity (Fig 2B). Such clones might still be reactive to SARS-CoV-2 antigens other than those tested. These clonotypes were also among the top 12 largest in size (Fig 2F). In particular, CoV1901/1919 and CoV1907/1912 lineages were the first and second largest lineages with 23.1% and 14.0% of PBs, respectively, in donor SG CO OK 00019 (Fig 2F). Clones reactive to both viral antigens and CL had more mutations than the rest of the antiviral clones without anti-CL reactivity (Fig 2G). Notably, all clones reactive to both CL and S were found to be highly mutated and cross-reactive to OC43 S (Table S2). Collectively, robust clonal expansion of highly mutated self-reactive B cells was observed during SARS-CoV-2 infection.

## Identification of a public autoantibody, PA-N-CoV1804 in COVID-19 patients

Although viral infections often induce convergent or public antibody responses, public clonotype repertoires of autoantibodies, defined as antibodies sharing genetic features and structural modes of antigen recognition across multiple individuals, are relatively uncommon. Taking advantage of the large sample size, we searched scRNAseq data of the Osaka cohort for clonotypes sharing genetic features with the anti-CL mAbs we isolated. We identified multiple COVID-19 patients possessing B-cell clones sharing genetic features with CoV1804 and CoV1838 that also reacted to CL and the nucleoli of HEp-2 cells (Table 1 and Fig 2D). A total of 340 of 50,755 COVID-19 patient–derived and 56 of 40,385 healthy donor–derived clones were encoded by a combination of *VH3-30/3-33* (*IGHV3-30/3-33*) and *Vλ4-69* (*IGLV4-69*) in the same manner as CoV1804 (Fig S7A and B). The heavy chain CDR3 (HCDR3) length of

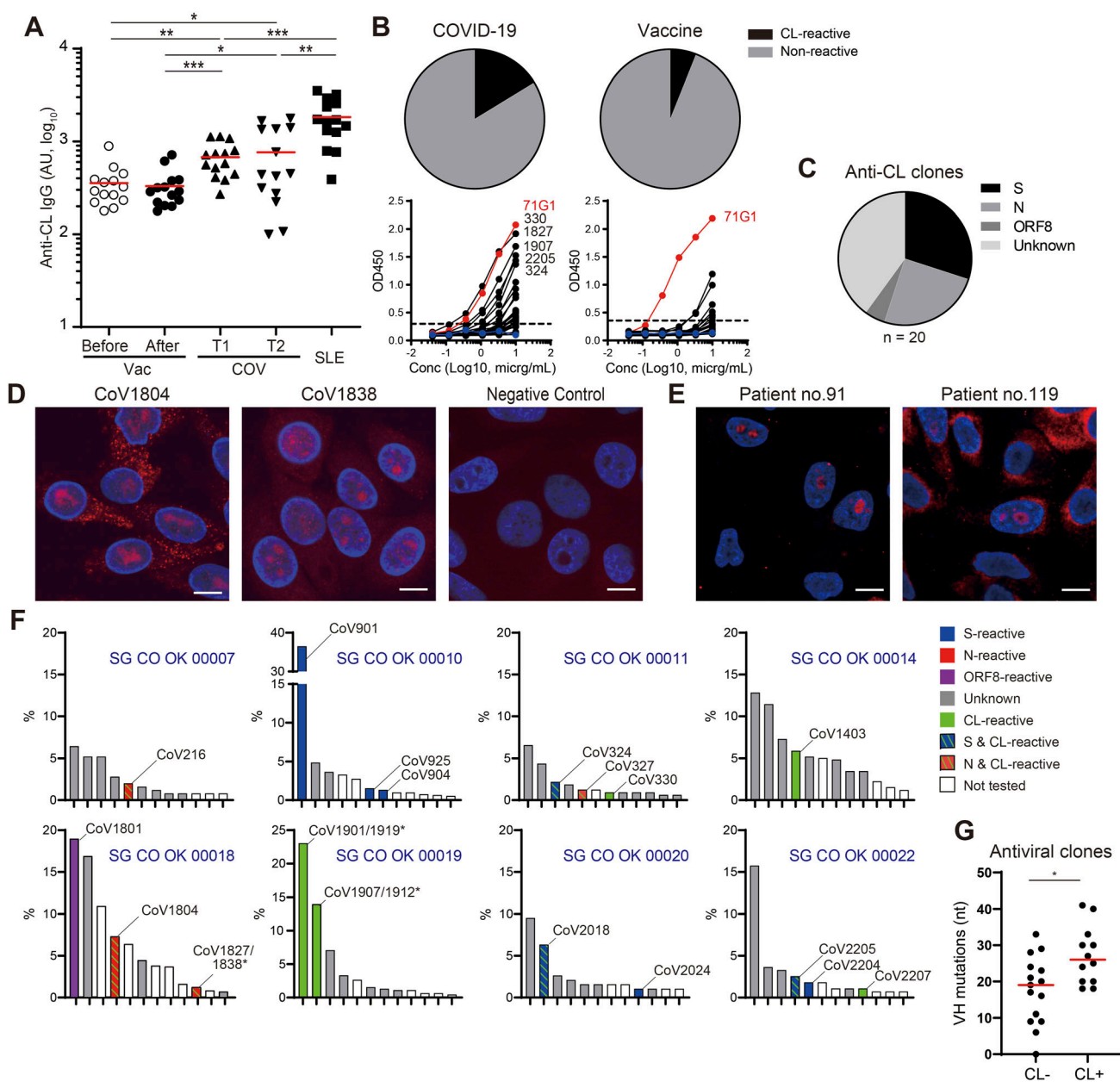

**Figure 2. Expansion of clones reactive to both viral and self-antigens in COVID-19 patients.**
**(A)** Plasma IgG antibody titers against cardiolipin (CL) (N = 14 each). AU: arbitrary unit. Mann–Whitney test: *≤0.05, **≤0.01, ***≤0.001 **(B)** Proportions of anti-CL clones in the infection- and vaccination-induced mAb panels (the upper pie charts). Representative ELISA results for CL reactivity of mAbs (the lower graphs). The dotted line indicates the threshold of positivity in each assay. **(C)** Antiviral reactivity of anti-CL mAbs from COVID-19 patients. **(D, E)** HEp-2 staining with (D) 1 μg/ml of mAb (red), or (E) 1: 200 diluted plasma (red). DAPI (blue). Bars = 10 μm. **(F)** Clonal size distribution of eight patient donors who had anti-CL clones in the largest clones. Each bar indicates the clonal proportion (%) in PBs. The colors indicate reactivity of each clonotype. * Related clones are combined. **(G)** VH mutations of antiviral mAbs with and without CL reactivity (n = 15 and 12, respectively). **(A, B, D, E)** Representative results from three independent assays are shown (A, B, D, E). Mann–Whitney test: *P ≤ 0.05. Source data are available for this figure.

clones having *VH3-30/3-33* and *Vλ4-69* from patients was found to be strictly constrained to 14 amino acids (Fig 3A). Clones encoded by *VH3-30/3-33* and *Vλ4-69*, carrying 14-aa-long HCDR3s amounted to 286 of 50,755 compared with 3 of 40,385 in COVID-19 and healthy donors, respectively (Fig S7A). Multiple sequence alignment analysis highlighted several residues such as E107, D110, and S113 conserved at the HCDR3 junction of CoV1804-like

clonotypes (Fig 3B). Moreover, the LCDR3 sequences from these clonotypes were all 9-aa-long with little diversity (Fig S7C).

The above genetic features of CoV1804-like clonotypes were also found to be shared by nCoV396, a previously reported anti-SARS-CoV-2 N mAb (Kang et al, 2021), which was consistent with the fact that CoV1804 and CoV1838 were reactive to not only CL but also to N protein (Fig 3C). Notably, E107, D110, S113, and W114 of the HCDR3

**Table 1. Infection-induced CL-reactive clones.**

| Donor ID | Clone ID | Isotype | IGHV | IGHJ | HCDR3 | VH mt | IGK/LV | IGK/LJ | LCDR3 | Vκ/VΛ mt | Size (%) | Reactivity |
|---|---|---|---|---|---|---|---|---|---|---|---|---|
| SG CO OK 00007 | CoV216 | IGHG1 | IGHV1-24 | IGHJ3 | CATEAVGPTVIYAFANW | 25 | IGKV3-20 | IGKJ1 | CQQYGSSPTF | 10 | 2.4 | N |
| SG CO OK 00010 | CoV915 | IGHG1 | IGHV3-7 | IGHJ6 | CARHAPSCNNGICYYLHYYMDVW | 30 | IGKV3-15 | IGKJ1 | CQQYHNWPPWTF | 16 | 0.11 | S |
| | CoV923 | IGHG1 | IGHV4-34 | IGHJ6 | CARGRLEWPAPILGLGPFYYSYYMDVW | 10 | IGLV3-19 | IGLJ2 | CNSRDSSGDHLVF | 10 | 0.11 | Unknown |
| SG CO OK 00011 | CoV324 | IGHG1 | IGHV3-30 | IGHJ3 | CVRPVELFWLGQFKRDAFDLW | 30 | IGLV1-44 | IGLJ2 | CAAWDDNLNGVVF | 18 | 2.2 | S |
| | CoV327 | IGHG3 | IGHV4-59 | IGHJ4 | CARGHMVGGTYYGLDFW | 40 | IGLV3-25 | IGLJ3 | CQSAGSNTHYQVF | 20 | 1.3 | N |
| | CoV3179 | IGHG1 | IGHV3-23 | IGHJ3 | CAKELSRGIRGIWGSNVFDVW | 18 | IGKV1-6 | IGKJ1 | CLQEYNYPRTF | 8 | 0.31 | S |
| | CoV330 | IGHG1 | IGHV4-61 | IGHJ6 | CARLKRAMVRGVKGLEIGMDVW | 8 | IGKV1-9 | IGKJ2 | CQQLNSYPLYTF | 10 | 0.93 | Unknown |
| | CoV1403 | IGHG1 | IGHV3-30 | IGHJ4 | CAKETEDSSSSFFDYW | 25 | IGLV4-69 | IGLJ3 | CQTWGSGLQVF | 15 | 5.2 | Unknown |
| SG CO OK 00014 | CoV1410 | IGHG1 | IGHV5-51 | IGHJ4 | CVRLGGINWGSINW | 20 | IGKV2D-40 | IGKJ4 | CLQRIAFPLTF | 6 | 0.52 | S |
| | CoV1427 | IGHA1 | IGHV3-23 | IGHJ6 | CVRHEFLGLTDIWRSDYYYGMEVW | 20 | IGLV1-47 | IGLJ3 | CASWDDSLSGWVF | 9 | 0.17 | ORF8 |
| SG CO OK 00018 | CoV1804 | IGHG1 | IGHV3-30 | IGHJ4 | CARECDDFNHSWFDYW | 27 | IGLV4-69 | IGLJ2 | CQTWGTGVQVF | 25 | 6.3 | N |
| | CoV1827 | IGHG1 | IGHV3-30 | IGHJ5 | CARETLDPGSSWFSHW | 41 | IGLV4-69 | IGLJ3 | CQTWATGIQVF | 26 | 0.65 | N |
| | CoV1838 | IGHG1 | IGHV3-30 | IGHJ5 | CARETPDPGSSWYSHW | 24 | IGLV4-69 | IGLJ3 | CQTWGTGIQVF | 15 | 0.32 | N |
| | CoV1901 | IGHA1 | IGHV7-4-1 | IGHJ3 | CARNHVVELTASGAFDVW | 28 | IGLV3-25 | IGLJ2 | CTSADSTNALVVF | 17 | 18 | Unknown |
| | CoV1907 | IGHG1 | IGHV2-70 | IGHJ6 | CAGTLDCGDDSYYYHGVDVW | 21 | IGLV6-57 | IGLJ2 | CQSYDSRSVVF | 12 | 5.8 | Unknown |
| SG CO OK 00019 | CoV1912 | IGHG1 | IGHV2-70 | IGHJ6 | CARTLDCGDDSYYYYGVDVW | 16 | IGLV6-57 | IGLJ2 | CQSYDSSSVVF | 6 | 4.0 | Unknown |
| | CoV1919 | IGHA1 | IGHV7-4-1 | IGHJ3 | CARNHVVDLTASGAFDVW | 27 | IGLV3-25 | IGLJ2 | CTSADGTNALVVF | 13 | 0.89 | Unknown |
| SG CO OK 00020 | CoV2018 | IGHG1 | IGHV3-23 | IGHJ4 | CAKDYPEMIRGVIIGERYFDYW | 18 | IGLV2-23 | IGLJ3 | CCSYVGNSFWVF | 24 | 4.7 | S |
| SG CO OK 00022 | CoV2205 | IGHG1 | IGHV3-33 | IGHJ3 | CAKTDTFFWLGEGRGVFDFW | 33 | IGKV1-17 | IGKJ4 | CLQHNSYPLTF | 14 | 1.1 | S |
| | CoV2207 | IGHA1 | IGHV5-51 | IGHJ3 | CARPRSGGYPAGDAFDIW | 22 | IGKV1-NL1 | IGKJ1 | CQQYFSSPGTF | 11 | 1.1 | Unknown |

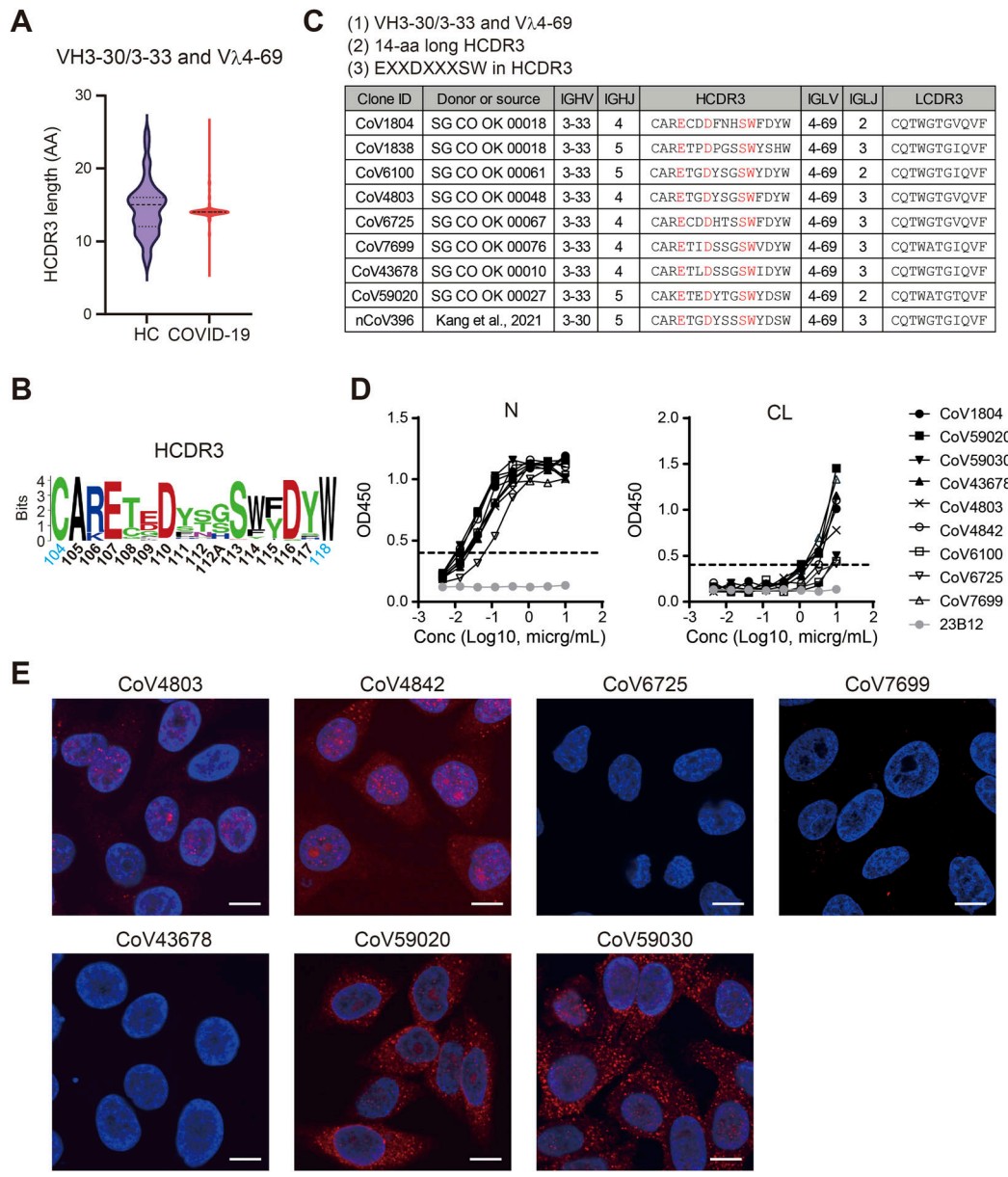

**Figure 3. Identification of a novel public antibody class in COVID-19–derived Igs.**
**(A)** The HCDR3 length of Igs encoded by *VH3-30* or *3-33* and *Vλ4-69* in the Osaka cohort. HC, n = 56; and COVID-19, n = 340. See also Fig S7. **(B)** Sequence logo of the CDR3 of 1804-like IgHs. **(C)** The representative clones belonging to PA-N-CoV1804. The EXXDXXXSW motif is colored in red. **(D)** Anti-N and anti-CL reactivity of the representative PA-N-CoV1804 antibodies in ELISA. The dotted line indicates the threshold of positivity in each assay. **(E)** HEp-2 staining with 1 μg/ml of mAb (red) and DAPI (blue). Bars = 10 μm. **(D, E)** Representative results from three independent assays are shown (D, E).
Source data are available for this figure.

were conserved and critically located in the paratope of nCoV396 (Kang et al, 2021) (Figs 3C and S7D). Also, the light chain CDR3 (LCDR3), which were shown to be crucial for antigen recognition (Kang et al, 2021), had high similarity to clones related to CoV1804 (Fig S7D). Based on these observations, we defined CoV1804-like public antibodies as PA-N-CoV1804 (public antibody against N, CoV1804-like) as follows: (1) The variable genes of heavy and light chains were *VH3-30/3-33* and *Vλ4-69*, respectively; (2) The CDR3 of the heavy chain was 14-aa long; and (3) contained the EXXDXXXSW motif (where X can be any amino acid). Using these criteria, we

found 213 sequences that belonged to PA-N-CoV1804 from 7 of 53 patients (13.2%), whereas no sequence among healthy donors belonged to this antibody class (Fig S7A). Consequently, these PA-N-CoV1804 sequences were encoded by various J segments, including *JH4* (*IGHJ4*) or *JH5* (*IGHJ5*) for the heavy chain, and *Jλ2* (*IGLJ2*) or *Jλ3* (*IGLJ3*) for the light chain (Fig 3C).

We expressed several PA-N-CoV1804 antibodies from representative patient donors in the Osaka cohort and confirmed their binding to N protein in ELISA (Fig 3D). Unlike the strong binding to N protein, the self-reactivity of PA-N-CoV1804 was divergent in ELISA,

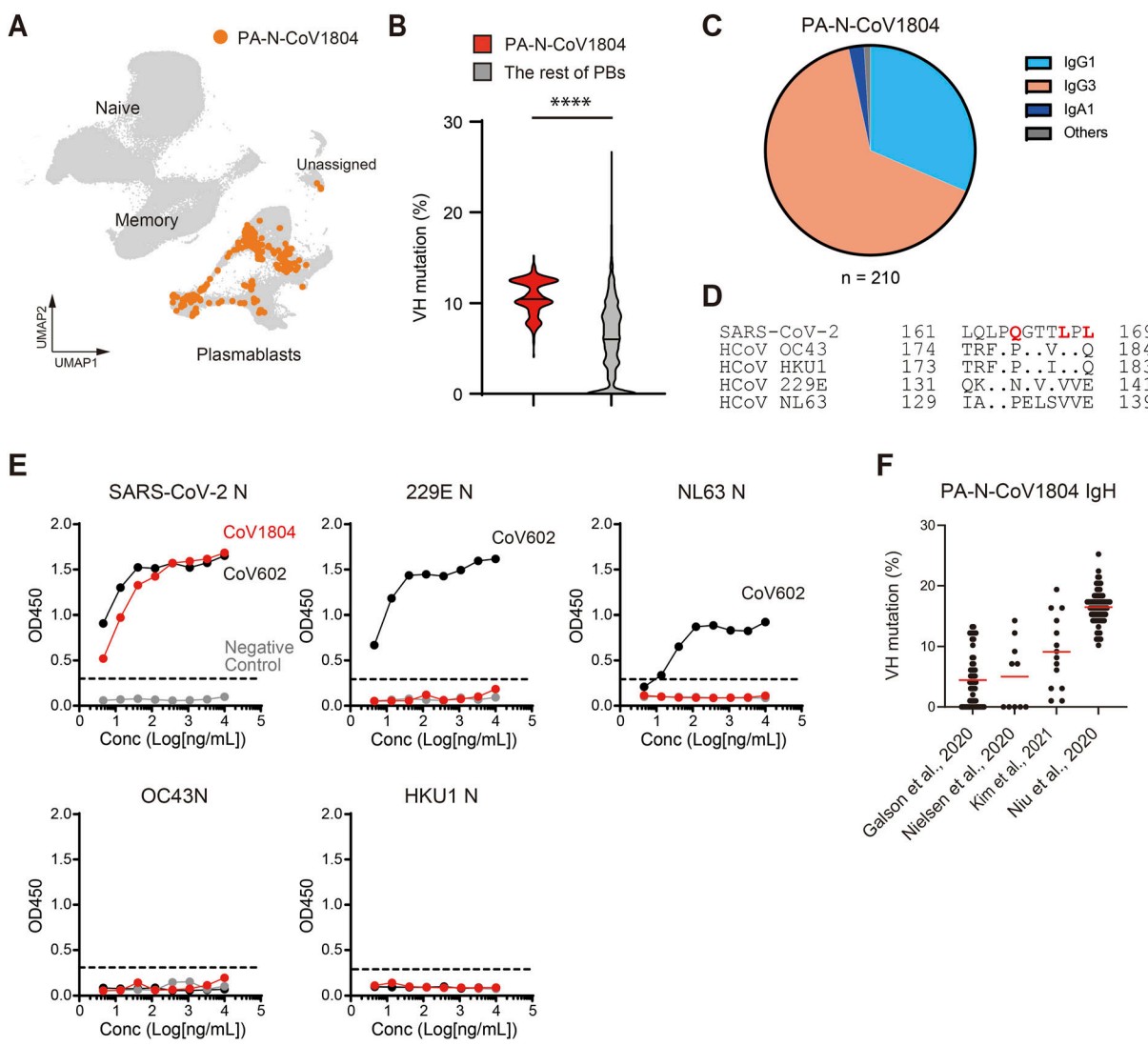

**Figure 4. PA-N-CoV1804 antibodies are de novo generated from naive B cells after SARS-CoV-2 infection.**
**(A)** PA-N-CoV1804-expressing cells in the UMAP plot of the Osaka cohort. **(B)** VH mutation rates of PA-N-CoV1804 and the rest of Igs expressed in PBs. PA-N-CoV1804, n = 210; and non-PA-N-CoV1804, n = 20622. Mann–Whitney test, two-tailed: ****$P ≤ 0.0001$. **(C)** The Ig classes of PA-N-CoV1804. **(D)** Amino acid sequence alignment of the region encompassing the epitope of PA-N-CoV1804 in coronavirus N proteins. **(E)** Reactivity of CoV1804 (red) and a cross-reactive clone, CoV602 (black) to N proteins from SARS-CoV-2 and seasonal coronaviruses in ELISA. The dotted line indicates the threshold of positivity in each assay. **(F)** VH mutation rates of PA-N-CoV1804 sequences from previously reported bulk Ig sequencing datasets (Galson et al, 2020) (Nielsen et al, 2020) (Niu et al, 2020; Kim et al, 2021). Red horizontal lines represent the mean of the group. **(E)** Representative results from three independent assays are shown (E).
Source data are available for this figure.

and only a subset of the tested antibodies, i.e., CoV4803, CoV4842, CoV59020, and CoV59030, exhibited ANoA activity upon HEp-2 cell staining (Fig 3D and E).

We further explored the presence of PA-N-CoV1804 in other published COVID-19 B-cell receptor (BCR) datasets (Bernardes et al, 2020; Galson et al, 2020; Kuri-Cervantes et al, 2020; Nielsen et al, 2020; Niu et al, 2020; Schultheiss et al, 2020; Montague et al, 2021 *Preprint*; Notarbartolo et al, 2021; Stephenson et al, 2021). In scRNAseq studies, which consisted of 149 patient donors in total, we found three COVID-19 patients who possessed at least one B-cell expressing PA-N-CoV1804 (Table S5). In bulk sequence datasets consisting of 55 million unique IgH sequences from 182 donors, 154

sequences from 13 patients with COVID-19 belonged to PA-N-CoV1804 (Table S6).

### PA-N-CoV1804 antibodies are de novo generated in response to SARS-CoV-2 infection but highly divergent from naive B-cell repertoire

In the Osaka scRNAseq data, the PA-N-CoV1804 Igs were detected exclusively in PBs (210/213 [98.6%]) (Fig 4A), indicating that they had been secreted as antibodies in these patients. The mutation rates in the PA-N-CoV1804 heavy chains were significantly higher than those expressed in other PBs (Fig 4B). Unlike anti-S2 cross-reactive

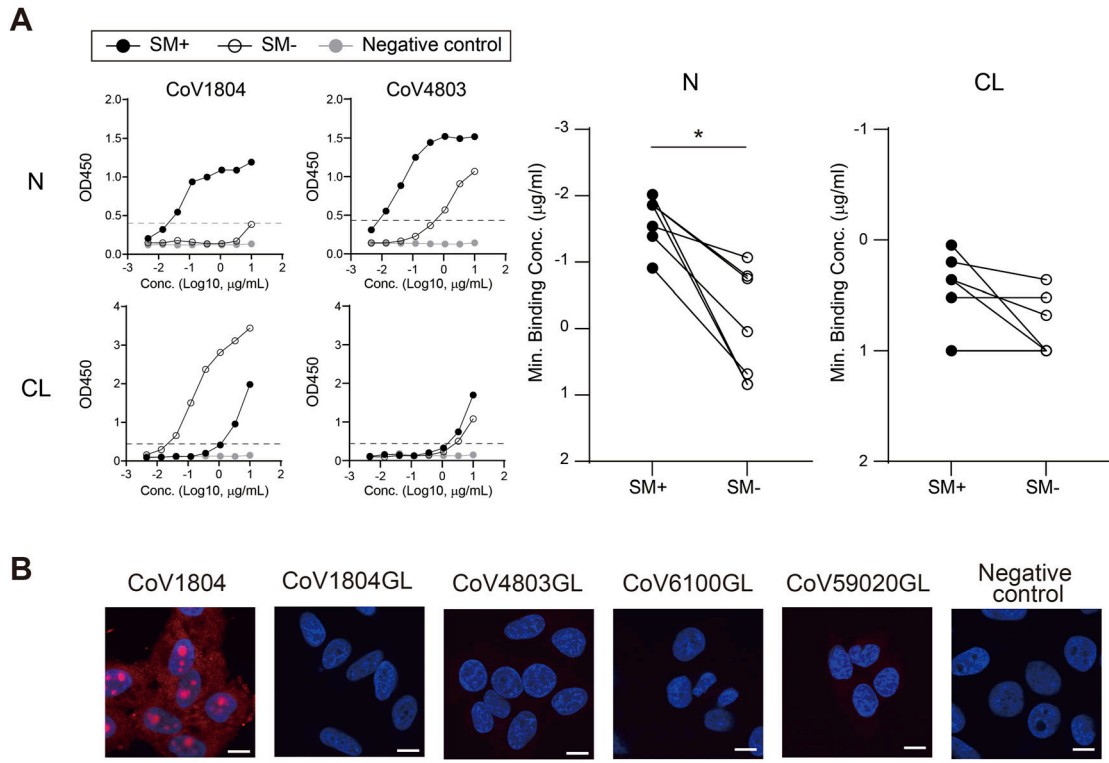

**Figure 5. SHM unambiguously enhances anti-N reactivity and is critical for anti-nucleolar activity but fluctuates anti-CL reactivity of PA-N-CoV1804.**
**(A)** N- and CL reactivities of mutated PA-N-CoV1804 (SM+) and the reverted germline antibodies (SM−) were assessed by ELISA. The representative results of CoV1804 and CoV4803 are shown on the left (the gray line represents the reactivity of negative control human IgG1. The dotted line indicates the threshold of positivity in each assay). Summary of three independent assays of CoV1804, CoV4803, CoV6100, CoV7699, CoV6725, CoV43678, and CoV59020 are plotted in the right panels. **(B)** HEp-2 staining with 1 μg/ml of the germline antibodies (mAb [red] and DAPI [blue]). Bars = 10 μm.
Source data are available for this figure.

clones in the initial screening, which were highly restricted to IgG1, the PA-N-CoV1804 clones in PBs of the Osaka cohort were diverse, including IgG1, IgG3, and IgA1 (Fig 4C). Owing to a lack of sequence similarity in local regions encompassing the targeted epitope among the N protein family (Kang et al, 2021), CoV1804 bound to N protein from SARS-CoV-2, but not seasonal coronaviruses (Fig 4D and E). Thus, the source of PA-N-CoV1804 antibodies may be de novo stimulated naïve B cells. This notion was further supported by bulk Ig analysis of COVID-19 patients from the public repository reported before 2022, in which three of four cohorts had not only mutated but also germline or near germline PA-N-CoV1804 sequences (Fig 4F). This indicates robust mutational diversification of the public antibodies after SARS-CoV-2 infection.

## Multiple self-reactivity of PA-N-CoV1804 is shaped by N protein-driven antibody response

From the above, PA-N-CoV1804 antibodies appeared to be derived from naïve B cells, yet accumulated somatic mutations (Fig 4B). To assess the importance of somatic hypermutation (SHM) in the maturation of PA-N-CoV1804 antibodies, we compared the reactivity of mutated and germline (GL) Ig-derived GL antibodies. All the tested clonotypes exhibited enhanced anti-N reactivity through SHM (Fig 5A). On the other hand, albeit the GL antibodies of PA-N-

CoV1804 already had some reactivity to CL, such self-reactivities were not always augmented in their descendants through SHM (Fig 5A). For example, the GL antibody of CoV1804 displayed strong anti-CL reactivity, which was more potent than mature CoV1804 (Fig 5A). These results suggested that anti-N reactivity may be the main driver for the development of PA-N-CoV1804 clonotypes. Whereas some of the PA-N-CoV1804 clones including CoV1804 showed anti-nucleolar activity in human cells (Fig 3E), all the GL clones tested showed no reactivity to HEp-2 cells (Fig 5B). Thus, SHM also played a role in the sporadic anti-nucleolar activity of PA-N-CoV1804.

We then investigated the phylogenetics and functional diversities of PA-N-CoV1804 in single donors. We found 26 unique sequences belonging to PA-N-CoV1804 in the PB-derived Igs of donor SG CO OK 00018. The sequences phylogenetically clustered into three clades, 1a, 2a, and 3a (Fig 6A). Given the diversity observed in HCDR3 between these clades (Fig 6B and Table S7), PA-N-CoV1804 antibodies in this donor may originate from several distinct germline precursors, which were reproducibly selected in antibody response to SARS-CoV-2 infection. All the clades consisted of somatically mutated clones (Fig 6B). Particularly, clade 3a, which included CoV1827 and CoV1838, had several clones with higher mutation rates (Fig 6C). All tested clones showed strong binding activity to N (minimal binding concentration ≤ 0.12 μg/ml) (Fig 6D). However, they had different potencies in binding to CL

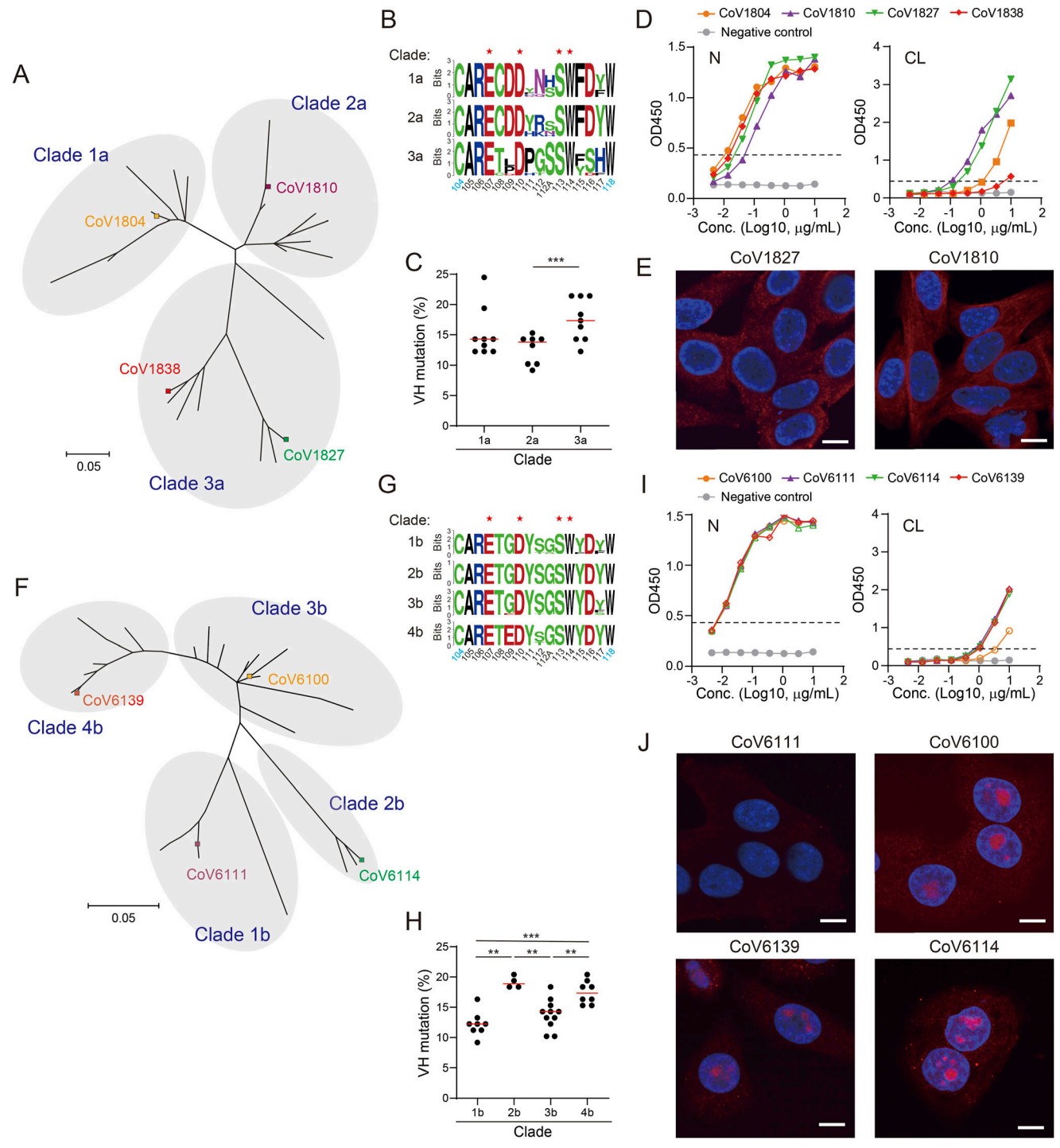

**Figure 6. Sequence and functional diversity of PA-N-CoV1804 in COVID-19 patients.**
**(A)** Phylogenic tree of IgH (VH-DH-JH, aa 1-132, IMGT numbering) of PA-N-CoV1804 derived from donor SG CO OK 00018. **(B)** Sequence logo of HCDR3. **(C)** VH mutation frequencies in individual phylogenetic clades. **(D)** Anti-N and anti-CL reactivity in ELISA. **(E)** HEp-2 cell staining of representative clones from individual clades. **(F)** Phylogeny of PA-N-CoV1804 IgH derived from donor SG CO OK 00061. **(G)** Sequence logo of HCDR3. **(H)** VH mutations in individual clades. **(I)** Anti-N and anti-CL reactivity in ELISA. **(J)** HEp-2 cell staining of representative clones from individual clades. **(A, F)** The maximal-likelihood model was used to construct the unrooted dendrograms (A, F). **(B, G)** Amino acids marked with asterisk (*) indicate the EXXDXXXSW motif (B, G). Red horizontal lines represent mean of the group. **(C, H)** Mann–Whitney test: **≤0.01, ***≤0.001 (C, H). **(D, I)** The dotted lines indicate the threshold of positivity in each assay (D, I). mAb (red) and DAPI (blue). Bars = 10 μm (E, J). **(D, E, I, J)** Representative results from three independent assays are shown (D, E, I, J).
Source data are available for this figure.

(Fig 6D). In HEp-2 staining assay, CoV1810 and CoV1827 were found to have anti-cytoplasmic activity, which was qualitatively distinct from the anti-nucleolar activity of CoV1804 and CoV1838 (Figs 2B and 6E).

The PA-N-CoV1804 sequences from another patient donor, SG CO OK 00061 also illustrated the variations in self-reactivity of this public antibody. We found 31 unique PA-N-CoV1804 sequences in this patient's BCR repertoire, which clustered into four clades, 1b, 2b, 3b, and 4b with highly similar HCDR3 sequences (Fig 6F and G). Most of the PA-N-CoV1804 sequences from this donor were encoded by *VH3-33/DH6-13/JH5* and *Vλ4-69/Jλ2* (Table S8), implying that they may originate from the same germline precursor. The PA-N-CoV1804 clones in this donor also had numerous somatic mutations, particularly in clades 2b and 4b (Fig 6H). Anti-N reactivity was uniformly potent across all tested PA-N-CoV1804 clones, whereas anti-CL reactivity and anti-nucleolar activity for HEp-2 cells were not consistently observed: CoV6100 showed lower anti-CL reactivity in ELISA, and CoV6111 showed no self-reactivity in HEp-2 cell staining assay (Fig 6I and J).

Overall, despite the inter- and intraclonal sequence diversity, PA-N-CoV1804 clones maintained strong anti-N reactivity. In contrast, this sequence diversity conferred the fluctuations in self-reactivities of PA-N-CoV1804, including anti-CL reactivities and anti-nucleolar activities within single donors.

### Somatic mutations contribute to the increased spectrum of self-reactivity of PA-N-CoV1804

Because some PA-N-CoV1804 clones exhibited both anti-CL reactivity and ANoA activity, we then conducted extensive investigation into the multiple reactivity of the representative clones against structurally different substances, including dsDNA, insulin and LPS. Whereas most PA-N-CoV1804 antibodies showed anti-dsDNA, they did not strongly react to insulin or LPS (Fig 7A). In particular, CoV1810, CoV1827, CoV59020, and CoV59030 showed strong anti-dsDNA reactivity, which was comparable with a previously isolated SLE-related ANA clone, 71F12 (Sakakibara et al, 2017) (Fig 7A). In addition to anti-dsDNA reactivity, many PA-N-CoV1804 clones exhibited diverse reactivities to multiple targets such as CL, other phospholipids, and the nucleolar and cytoplasmic antigens in HEp-2 cells (Fig 7B). Interestingly, the number of targets recognized by PA-N-CoV1804 clones was correlated with the extent of somatic mutation in their VH segments (Fig 7C). Thus, the accumulation of somatic mutations appears to increase the spectrum of self-reactivity of PA-N-CoV1804.

## Discussion

In this study, we investigated the nature of antibody responses during severe COVID-19. Although not definitive, antibody response in COVID-19 ICU patients appeared to be a mixture of primary and recall responses. SARS-CoV-2 infection strongly reactivates pre-existing B-cell memory for seasonal coronaviruses, in line with the notion of "original antigenic sin" that has long been described in other viral infections (Angeletti et al, 2017; Zhang et al, 2019). In a panel of mAbs derived from COVID-19 patients, we frequently found

cross-reactive anti-S antibodies, which were IgG1 and carried a relatively high number of mutations but lacked neutralizing activity. A typical primary antibody response also occurs in COVID-19 ICU patients, generating specific antiviral antibodies encoded by a variety of Ig classes and with a wide range of somatic mutations. Of these specific antibodies, anti-N or anti-ORF8 clones were more mutated than anti-S clones, possibly because the kinetics of antibody response differ for each target. Particularly, SARS-CoV-2 S-specific neutralizing antibodies may begin to expand only later in infection. In addition, robust clonal expansion of autoantibody-secreting plasmablasts occurs in severe COVID-19 patients. Half of these autoantibodies also reacted with S, N, or ORF8 of SARS-CoV-2. These autoantibodies contained a high number of mutations. All the S-reactive autoantibody clones showed cross-reactivity to SARS-CoV-2 and OC43 S, suggesting that they originated from pre-existing memory B cells, whereas those reacting to ORF8 or N without anti-seasonal coronavirus reactivity appeared to originate from primary responses to SARS-CoV-2. Assuming that the number of mutations roughly reflects the number of cell divisions, early antibody responses in severe COVID-19 may be dominated by not only antibodies derived from recall responses to conserved coronaviral antigens but also those from primary responses against SARS-CoV-2, some of which are self-reactive.

A recent study have reported that COVID-19–associated autoantibody potentially originated directly from naive B-cell precursors, with identification of unmutated IgG1 clones reactive to both COVID-19 and self-antigens (Woodruff et al, 2020; Woodruff et al, 2022). However, we did not observe such clones in a panel of mAbs derived from COVID-19 patients. Instead, autoantibodies isolated in this study were highly mutated, indicating that COVID-19–associated autoantibodies may be the result of clonal expansion and SHM. Thus, our findings suggest another possible origin of infection-induced autoantibodies.

From these autoantibodies, we identified the public clonotype, PA-N-CoV1804, which reacts to SARS-CoV-2 N in a diverse set of donors. This provided a novel finding on the functional diversity of autoantibodies with oligoclonal antibody convergence and subsequent SHM in antiviral responses. Despite their anti-N reactive GL precursors, the PA-N-CoV1804 sequences underwent progressive mutational diversification, in which SHM readily enhanced the reactivity to SARS-CoV-2 N. This suggests that PA-N-CoV1804 antibodies were rigorously selected based on their reactivity to N protein. During this antibody response, PA-N-CoV1804 clones broadened their spectrum of target self-antigens but did not consistently enhance reactivity to any particular self-antigen. Consequently, the sequence diversity of PA-N-CoV1804 conferred different patterns of self-reactivity in each phylogenetic path.

The multiple self-reactivity or polyreactivity of PA-N-CoV1804 clones was similar to the reactivity that has been observed in antibodies derived from human and mouse CD5[+] B cells (i.e., B-1 cells) (Casali & Notkins, 1989). The proposed function of polyreactive antibodies was as a first line of defense against a wide variety of pathogens (Ochsenbein et al, 1999; Zhou et al, 2007; Gronwall et al, 2012; Gunti & Notkins, 2015). Polyreactive antibodies also bind to apoptotic cells and facilitate their clearance (Gunti & Notkins, 2015). Therefore, CL-reactive PA-N-CoV1804 producing B cells may have a competitive advantage over virus-specific clones

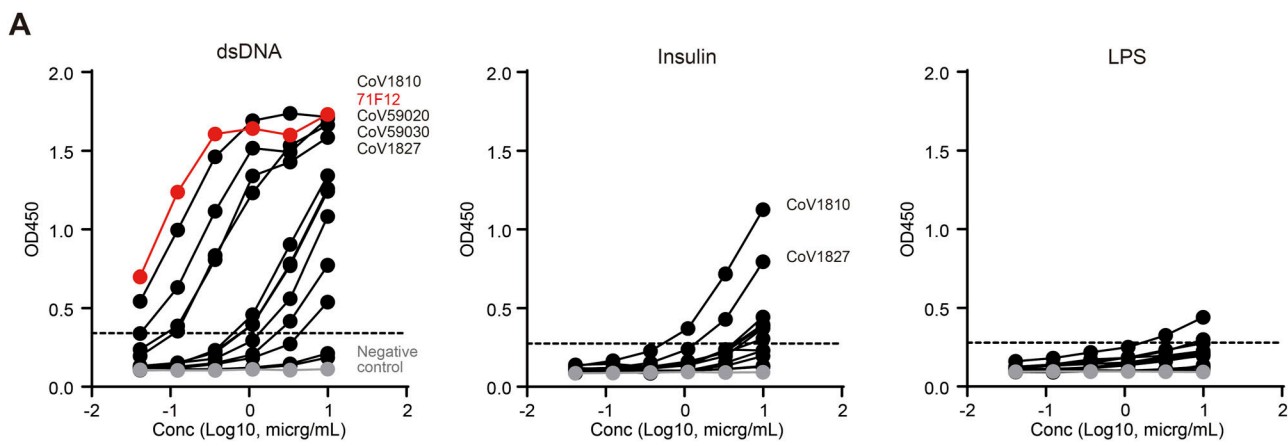

**A**

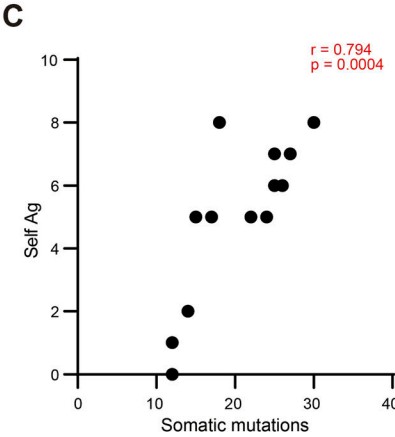

**B**

| Donor ID | Clone ID | CL | PA | PC | PE | dsDNA | LPS | insulin | Hep2 | #Ag | SHM |
|---|---|---|---|---|---|---|---|---|---|---|---|
| SG CO OK 00018 | CoV1804 | + | + | + | + | - | - | - | +, Ncl | 5 | 14 |
| | CoV1810 | +++ | ++ | ++ | ++ | ++ | + | + | +, Cyt | 8 | 12 |
| | CoV1827 | ++ | ++ | + | + | ++ | + | + | +, Cyt | 8 | 22 |
| | CoV1838 | ++ | + | + | + | - | - | - | +, Ncl | 5 | 13 |
| SG CO OK 00061 | CoV6100 | + | - | - | - | - | - | - | +, Ncl | 2 | 12 |
| | CoV6111 | + | + | + | + | + | - | + | - | 6 | 15 |
| | CoV6114 | + | + | + | + | + | - | + | +, Ncl | 7 | 20 |
| | CoV6139 | ++ | + | + | + | + | - | + | +, Ncl | 7 | 20 |
| SG CO OK 00027 | CoV59020 | + | ++ | + | + | +++ | - | + | +, Cyt | 7 | 20 |
| | CoV59030 | + | + | + | + | ++ | - | - | +, Cyt | 6 | 20 |
| SG CO OK 00048 | CoV4803 | + | + | + | + | + | - | - | - | 5 | 18 |
| | CoV4842 | + | + | + | + | - | - | - | +, Ncl | 5 | 18 |
| SG CO OK 00010 | CoV43678 | - | - | - | - | - | - | - | - | 0 | 11 |
| SG CO OK 00067 | CoV6725 | + | - | - | - | - | - | - | - | 1 | 10 |
| SG CO OK 00070 | CoV7699 | + | + | + | - | + | - | + | - | 5 | 16 |

**C**

r = 0.794
p = 0.0004

**Figure 7. SHM increases a spectrum of self-reactivity of PA-N-CoV1804 antibodies.**
**(A)** Multiple reactivity assay. The red line represents the reactivity of the 71F12 mAb (SLE patient-derived anti-dsDNA clone), and the gray line represents the reactivity of the negative control human IgG1. The dotted line indicates the threshold of positivity in each assay. **(B)** Reactivity to structurally unrelated substances and the heavy chain mutations of PA-N-CoV1804 antibodies. PA, phosphatidic acid; PC, phosphatidylcholine; PE, phosphatidylethanolamine; LPS, lipopolysaccharide; +, minimal concentration of 1–10 μg/ml; ++, 0.1–1 μg/ml; +++, 0.01–0.1 μg/ml; Ncl, nucleolar pattern; Cyto, cytoplasmic pattern. **(C)** Correlation between amino acid mutations in heavy chain and the number of targeted antigens. Spearman rank correlation and the *P*-value are shown.
Source data are available for this figure.

in the conditions where various cellular substances are released from tissues damaged by acute SARS-CoV-2 infection, but not mRNA vaccination. Under such conditions, additional reactivities against self-antigens such as nucleoli and DNA appear to be generated during mutational diversification alongside clonal expansion. Indeed, CL-reactive, antiviral clones had more mutations than antiviral clones not reacting to CL, suggesting these polyreactive clones may have expanded before virus-specific B cells.

We also found that several PA-N-CoV1804 antibodies exhibited potent anti-dsDNA reactivity and characteristic ANoA activity that resemble autoantibodies observed in patients with autoimmune diseases (Pinnas et al, 1973; Reveille et al, 2003; Rahman & Isenberg, 2008). This was interesting because some studies have shown that SARS-CoV-2 induced potentially pathogenic autoantibodies (Zuo et al, 2020; Chang et al, 2021; Gomes et al, 2021; Wong et al, 2021; Woodruff et al, 2022). Whether infection-induced autoantibodies contribute to the severity or specific symptoms including sequelae awaits further investigation. Another important finding in this study was that some of the isolated autoantibodies also reacted with viral antigens of seasonal coronaviruses. Therefore, not only SARS-CoV-2 but also other HCoVs such as those causing the common colds might contribute to the development of polyreactive B cells, which have been frequently observed in the IgG$^+$ memory B-cell compartment of healthy individuals (Tiller et al, 2007). In this context, a subset of self-reactive clones elicited by SARS-CoV-2 infection may be retained in the memory compartment and undergo reactivation upon re-exposure of the same or related viruses; such a phenomenon may contribute to some symptoms of post-COVID.

In our ICU patients, the association of PA-N-CoV1804 with disease symptoms was not found (data not shown). This was probably because our cohort included only ICU patients. Alternatively, not only the clone sizes but also mutation frequencies of PA-N-CoV1804 may be linked to autoantibody titers of COVID-19 patients. Antibody clonal landscape with a wide range of disease severities and symptoms, including post-acute COVID-19 sequelae, needs to be investigated in the future. Because of the nature of the disease, many of our COVID-19 ICU patients were elderly; clonal expansion of autoantibodies might thus be attributable to age-related alterations in immune function (Hijmans et al, 1984; Gibson et al, 2009). To address this point, age-stratified characterization of antibody response in COVID-19 patients would be warranted.

In summary, our results revealed the clonal landscape of autoantibody development during severe COVID-19. We found that broad self-reactivity arises from potentially CL-reactive precursors through viral antigen-driven clonal selection and expansion. This could be because of the nature of humoral response during viral infections, rather than an example of molecular mimicry or bystander polyclonal activation. Our study may also have implications for the process of autoantibody development associated with other viral infections, which, in term, may be a crucial trigger for autoimmune diseases.

# Materials and Methods

### Clinical subjects

Clinical samples were collected from COVID-19 patients admitted to the ICU of Osaka University Hospital (Osaka, Japan), between July 2020 and May 2021. COVID-19 ICU patients were classified as moderate and severe based on the World Health Organization (WHO) classification (WHO Working Group on the Clinical Characterisation and Management of COVID-19 infection, 2020). These patients received intensive care, including mechanical ventilation, anticoagulation therapy, and corticosteroids, in the ICU.

Blood samples were obtained on the first day of ICU admission (T1) and after treatment (days 7–22, T2). In addition, blood samples were collected from healthy volunteers who received the Pfizer-BioNTech SARS-CoV-2 mRNA vaccine (BNT162b2) before vaccination and 7–16 d after the second vaccination. Peripheral blood mononuclear cells (PBMCs) were isolated from the blood samples using a Ficoll gradient, frozen in CellBanker (Takara), and stored in the vapor phase of liquid nitrogen. Plasma samples were stored at –80°C until use.

### scRNAseq for PBMCs from vaccinated individuals

Frozen PBMCs were rapidly thawed in a 37°C water bath and subjected to droplet-based single-cell RNAseq using the Chromium Single Cell V(D)J Reagent Kits v1.1 (10x Genomics) according to the manufacturer's instructions. TCR and BCR V(D)J libraries, along with gene expression libraries, were generated for each sample. Library quality was evaluated using the Agilent bioanalyzer, and sequencing was performed on an Illumina NovaSeq 6000 using paired-end mode with a read1 length of 26 bp and a read2 length of 91 bp.

### Ig repertoire analysis of COVID-19 patients and healthy individuals

For this study, we reanalyzed scRNAseq data from COVID-19 patients and healthy controls collected at Osaka University Hospital (Edahiro et al, 2023), which is available in the DDBJ Japanese Genotype-phenotype Archive (JGA), accession number JGAS000543; JGAD000662; JGAS000593; JGAD000772. To ensure that the analysis was not influenced by vaccination or reinfection, we used sample data from 53 COVID-19 patients and 51 healthy donors collected between July 2020 and January 2021. Disease severity of COVID-19 patients was classified according to WHO guidelines. The raw Fastq files were processed using the 10x Genomics Cell Ranger 5.0 software with default settings, and transcriptome analysis was performed using Scanpy 1.8.11. To analyze B-cell clonotypes, we integrated clonotype information with the gene expression profiles of the scRNASeq data using Scirpy (Sturm et al, 2020) and excluded ribosome genes, *TRAV*, *TRAJ*, *TRBJ*, *TRBV*, *IGHV*, *IGKV*, and *IGLV* genes to avoid potential clustering bias.

We used Scrublet to predict and filter potential doublets (Wolf et al, 2018), and applied mixture of two Gaussian distributions to filter cell barcodes based on read counts, gene counts, and gene concentrations of ribosome, mitochondria, and hemoglobin genes (Wolock et al, 2019). Distribution of the read counts, count of gene per barcode, ribosome gene concentration, mitochondria gene concentration and hemoglobin gene concentration within the sample were each fitted with mixture of two Gaussian distributions. Only barcodes with IGH+IGK or IGH+IGL were retained for downstream analysis. The top 4,000 variable genes were selected using Scanpy (Wolf et al, 2018) and conducted batch effect correction using BBKNN (Polanski et al, 2020). We then performed Leiden clustering and PAGA graph analysis with Reingold-Tilford layout, integrated with UMAP projection (Buchheim et al, 2006; Becht et al, 2018; Traag et al, 2019; Wolf et al, 2019). The dataset was further processed using Dandelion and Immcantation for mutation rate

calling (Vander Heiden et al, 2014; Gupta et al, 2015; Stephenson et al, 2021).

## Cell lines

Expi293F cells (Thermo Fisher Scientific) were cultured in Expi293 Expression Medium (Gibco) in an incubator with humidified atmosphere of 8% CO2 in air on an orbital shaker rotating at 125 rpm ($1.74g$) at 37°C. HEK293T (ATCC), Vero E6 TMPRSS2 (JCRB1819; NIBION), and HEp-2 (ATCC) were cultured in DMEM supplemented with 10% FBS, 1 mM sodium pyruvate, 100 U/ml of penicillin–streptomycin in an incubator with humidified atmosphere of 5% $CO_2$ in air at 37°C.

## Ig cloning and expression

We isolated expanded PB-derived Igs from 12 randomly selected COVID-19 patients from the Osaka cohort. We prioritized largest clones in the respective subjects, except for those that failed to yield a sufficient amount of recombinant antibody. For patients with limited expansion, we randomly added several singletons. The Ig variable regions of the selected clonotypes were custom synthesized by IDT and Eurofins and inserted into mammalian expression vectors for human IgG1, Igκ, or Igλ. These single-cell-derived mAbs were then expressed and purified as previously reported (Shinnakasu et al, 2021). For clones reactive to multiple antigens, antibody size and purity were assessed by SDS–PAGE and size exclusion chromatography using AKTA pure 25 (GE Healthcare) equipped with Superose 6 Increase 10/300 GL (Cytiva). A total of 123 mAbs were generated from three patients with moderate symptoms and nine with severe symptoms. In addition, we generated 67 mAbs from PBs of three healthy individuals who received the Pfizer-BioNTech SARS-CoV-2 mRNA vaccine.

## Antigens

The source of antigens used for ELISA are listed in Fig S7. Recombinant SARS-CoV-2 S ectodomain trimeric form, recombinant RBD of S, and human coronavirus OC43 S ectodomain trimeric form were expressed as previously described (Shinnakasu et al, 2021). Briefly, supernatants from transfected cells were harvested 3 d after transfection, passed through 0.45-micron syringe filter (Millipore). The recombinant proteins were purified using Talon metal affinity resin (Takara), followed by buffer-exchanged into PBS and concentrated using Corning Spin-X centrifugal filters.

## ELISA

ELISA was performed as described elsewhere (El Hussien et al, 2022). mAbs were screened for reactivity to a panel of the following viral antigens: SARS-CoV-2 S trimer (154754; Addgene), S1 (40591-V08H; SinoBio), S2 (40590-V08B; SinoBio), N (230-01104-100; RayBiotech), and ORF8 (Bioworld Technology), OC43 S (VG40607-CF; SinoBio) and N (40643-V07E; SinoBio), 299E N (40640-V07E; SinoBio), NL63 N (40641-V07E; SinoBio) and HKU1 N (40642-V07E; SinoBio), and other substances: cardiolipin (CL, Sigma-Aldrich), bovine DNA (Sigma-Aldrich), bovine insulin

(Sigma-Aldrich), LPS (Sigma-Aldrich), Phosphatidic acid (PA, Avanti), Phosphatidylcholine (PC, Avanti), and Phosphatidyl-ethanolamine (PE, Avanti), using ELISA. Nunc Maxisorp ELISA plates were first coated with 50 ng of respective antigens in 50 μl of PBS and incubated overnight. The plates were washed once with PBS containing 0.1% Tween 20 (PBST) and then blocked with PBS containing 1% BSA (Nacalai Tesque) for 1 h at room temperature. Plasma samples or mAbs, which were serially diluted in 0.1% BSA-containing PBS were added to the plates and incubated at room temperature for 1 h. The plates were then washed thrice with PBST, and horseradish peroxidase-conjugated goat anti-human IgG (1:4,000 dilution; Southern Biotech) in PBST was added and incubated for 1 h at room temperature. The wells were washed thrice with PBST before the addition of 3,3′,5,5′-tetra-methylbenzidine substrate solution (Thermo Fisher Scientific). The reaction was stopped after 3 min by addition of 0.08 M sulfuric acid. The optical density at 450 nm was measured with a Promega GloMax Discover microplate reader. Clones that exhibited no reactivity to any the tested antigens or showed binding to BSA-coated wells, which served as control, were classified as having "unknown" reactivity.

For anti-phospholipid reactivity tests, 300 ng of respective phospholipids (CL, PA, PC, and PE) in 30 μl of ethanol was incubated on each well of Nunc Maxisorp ELISA plates until the solvent was vaporized at room temperature. The plates were then washed once with PBS and blocked with PBS containing 1% BSA for 1 h at room temperature. Plasma samples or mAbs serially diluted in 0.1% BSA-containing PBS were added to the plates and incubated at 37°C for 45 min. The plates were then washed thrice with PBS, and horseradish peroxidase-conjugated goat anti-human IgG (1: 4,000 dilution; Southern Biotech) in PBS containing 1% BSA was added and incubated for 1 h at room temperature. Reaction and measurement of OD450 were performed as described above. A SLE patient-derived anti-phospholipid autoantibody clone 71G1 was used for positive control (Sakakibara et al, 2017). An irrelevant antibody clone, 23B12 (Haruna et al, 2022) was used for negative control human IgG1 antibody. The threshold OD450 below which samples were considered negative was three times the value of that of the negative control.

## HEp-2 staining

For ANA staining, antibodies (≤1 μg/ml) were tested in indirect immunofluorescent assay with 8-chamber glass slides (Matsunami) containing HEp-2 cells (El Hussien et al, 2022) with minor modifications. HEp-2 cells were seeded on to the glass slides (Matsunami) and maintained overnight in DMEM supplemented with 10% FBS. After overnight cultivation, the cells were washed with PBS, fixed with 4% PFA in PBS and permeabilized by 0.1% Triton-X and 0.1% Tween 20. The fixed cells were then incubated with antibodies in 0.1% BSA/PBS for 30 min at 37°C, followed by three washes with PBS with 0.1% Tween 20. Bound antibodies were detected by anti-human IgG-Alexa Fluor 594 antibody (Thermo Fisher Scientific). The stained cells were pictured using an Olympus laser confocal microscope FV1000 equipped with a ×60 objective lens using the Fluoview Viewer software (Olympus).

### Pseudovirus neutralizing assay

Neutralizing assay for recombinant mAbs and plasma samples was performed as previously reported (Shinnakasu et al, 2021). Recombinant mAbs were serially diluted with DMEM containing 10% FBS to a concentration of 30 µg/ml. Heat-inactivated human plasma samples were serially diluted starting from 1/20 in culture medium. SARS CoV-2 S (Wuhan)–pseudotyped VSVΔG-luc was incubated with different dilutions of human plasma or recombinant antibodies for 1 h at 4°C and then inoculated onto a monolayer culture of VeroE6-TMRPSS2 in 96-well plates. After 16 h of incubation, cells were washed with PBS and lysed with luciferase cell culture lysis reagent (Promega). The cleared cell lysates were then incubated with firefly luciferase assay substrate (Promega) in 96-well white polystyrene plates (Corning). The luciferase activity was measured by a GloMax Discover luminometer (Promega). The NT50 or IC50 was calculated using Prism 9 software version 9.5.1 (GraphPad).

### Public antibodies

To identify a public antibody clonotype, PA-N-CoV1804, we conducted a search for BCR repertoire in scRNAseq datasets using specific criteria. We looked for sequences that contained either *VH3-30* or *VH3-33* and a 14-amino acid long HCDR3 (between C104 and W118) with the motif EXXDXXXSW (where X is any amino acid), paired with a *Vλ4-69* light chain. We excluded cells with multiple Ig chains from the analysis. We then expressed representative clonotypes as recombinant mAbs and tested their binding to the corresponding antigens.

### Phylogenetic analysis of PA-N-CoV1804 clonotype

To investigate the clonal evolution of PA-N-CoV1804 clonotypes, a maximum likelihood phylogenetic analysis was performed for the heavy chain variable regions using Mega-X (http://megasoftware.net). We analyzed unique sequences within the representative lineages to better understand the diversity and relatedness of the clonotypes.

### Statistics

Statistical analysis was performed using the Mann–Whitney *U* test (two-tailed) for comparison. *P*-values less than 0.05 were considered statistically significant. For correlation analysis, Spearman coefficients and *P*-values were calculated, with $P < 0.05$ considered statistically significant.

### Study approval

Clinical sample collection was conducted with approval from the Osaka University Research Ethics Committee. For the collection of bold samples, written informed consent was obtained from ICU patients or their relatives and healthy volunteers.

## Data Availability

Raw scRNAseq data of healthy individuals who received COVID-19 vaccine were deposited and available at Gene Expression Omnibus (GEO), accession number GSE233522. Source data are provided with this paper.

### Code availability

The Clonotype code used to determine the PA-N-CoV1804 is in the Gitlab at https://gitlab.com/sysimm/clonotype.

## Supplementary Information

## Acknowledgements

We thank all members of the All Handai Project and anonymous blood donors. We also thank Nana Iwami, Mayumi Yoshimoto (iFReC), the Core Instrument Facility (RIMD/IFReC) for their assistance in this work. This work was supported by JSPS KAKENHI (Y Okada, 22H00476), the Japan Agency for Medical Research and Development (AMED) (Y Okada, JP22km0405211, JP22ek0410075, JP22km0405217, JP22ek0109594, JP223fa627002, JP223fa627010, JP233fa627011, and JP23zf0127008; D Okuzaki, JP20fk0108404h0001), JST Moonshot R&D (Y Okada, JPMJMS2021, and JPMJMS2024), Takeda Science Foundation, Bioinformatics Initiative of Osaka University Graduate School of Medicine (Y Okada), Institute for Open and Transdisciplinary Research Initiatives, Center for Infectious Disease Education and Research (CiDER) (Y Okada), and Center for Advanced Modality and DDS (CAMaD), Osaka University (Y Okada), Kishimoto Foundation (H Kikutani), AMED-CREST (D Okuzaki, JP22gm1810003h0001), the Mitsubishi Foundation (D Okuzaki), and the NIPPON Foundation for Social Innovation (D Okuzaki).

### Author Contributions

S Sakakibara: conceptualization, validation, investigation, visualization, and writing—original draft, review, and editing.
Y-C Liu: data curation, formal analysis, validation, visualization, and writing—original draft, review, and editing.
M Ishikawa: data curation, investigation, and writing—review and editing.
R Edahiro: resources and writing—review and editing.
Y Shirai: resources and writing—review and editing.
S Haruna: investigation and writing—review and editing.
MA El Hussien: investigation and writing—review and editing.
Z Xu: data curation, formal analysis, validation, and writing—review and editing.
S Li: conceptualization, formal analysis, validation, and writing—review and editing.
Y Yamaguchi: resources and writing—review and editing.
T Murakami: resources and writing—review and editing.
T Morita: resources and writing—review and editing.
Y Kato: resources and writing—review and editing.
H Hirata: resources and writing—review and editing.
Y Takeda: resources and writing—review and editing.

F Sugihara: resources, validation, and writing—review and editing.

Y Naito: validation and writing—review and editing.

D Motooka: data curation, validation, and writing—review and editing.

C-Y Tsai: writing—review and editing.

C Ono: resources, methodology, and writing—review and editing.

Y Matsuura: resources, methodology, and writing—review and editing.

JB Wing: writing—review and editing.

H Matsumoto: resources, validation, and writing—review and editing.

H Ogura: resources and writing—review and editing.

M Okada: validation and writing—review and editing.

A Kumanogoh: resources and writing—review and editing.

Y Okada: resources, funding acquisition, and writing—review and editing.

DM Standley: data curation, formal analysis, validation, and writing—original draft, review, and editing.

H Kikutani: supervision, funding acquisition, and writing—original draft, review, and editing.

D Okuzaki: funding acquisition, project administration, and writing—review and editing.

## Conflict of Interest Statement

The authors declare that they have no conflict of interest.

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
