## [Reviewer comments · Life Science Alliance]

Life Science Alliance

Clonal landscape of autoantibody-secreting plasmablasts in COVID-19 patients

Shuhei Sakakibara, Yu-Chen Liu, Masakazu Ishikawa, Ryuya Edahiro, Yuya Shirai, Soichiro Haruna, Marwa El Hussien, Zichang Xu, Songling Li, Yuta Yamaguchi, Teruaki Murakami, Takeyoshi Morita, Yasuhiro Kato, Haruhiko Hirata, Yoshito Takeda, Fuminori Sugihara, Yuko Naito, Daisuke MOTOOKA, Chao-Yuan Tsai, Chikako Ono, Yoshiharu Matsuura, James Wing, Hisatake Matsumoto, Hisashi Ogura, Masato Okada, Atsushi Kumanogoh, Yukinari Okada, Daron Standley, Hitoshi Kikutani, and Daisuke Okuzaki

DOI: <https://doi.org/10.26508/lsa.202402774>

Corresponding author(s): Shuhei Sakakibara, Osaka University

Review Timeline:

Submission Date:	2024-04-17
Editorial Decision:	2024-08-19
Revision Received:	2024-08-28
Editorial Decision:	2024-09-04
Revision Received:	2024-09-05
Accepted:	2024-09-06

Transaction Report:

August 19, 2024

Re: Life Science Alliance manuscript #LSA-2024-02774

Dr. Shuhei Sakakibara
Osaka University
Immunology Frontier Research Center
3-1 Yamada-Oka
Suita, Osaka 565-0875
Japan

Dear Dr. Sakakibara,

Thank you for submitting your manuscript entitled "Clonal landscape of autoantibody-secreting plasmablasts in COVID-19 patients" to Life Science Alliance. The manuscript was assessed by expert reviewers, whose comments are appended to this letter. We invite you to submit a revised manuscript addressing the Reviewer comments.

Thank you for this interesting contribution to Life Science Alliance. We are looking forward to receiving your revised manuscript.

Sincerely,

B. MANUSCRIPT ORGANIZATION AND FORMATTING:

Reviewer #1 (Comments to the Authors (Required)):

This manuscript by Sakakibara et al. comprehensively describes clonal composition and diversity of SARS-CoV-2 reactive and cardiolipin reactive autoantibodies detected in severely ill patients. This is a very well-written and manuscript brings forth several important findings that are supported by the data. A major finding is description of a novel public antibody that recognise both viral and self antigen. Collectively this study provides fundamentally important insight into infection-associated autoimmunity. Only minor edits are suggested to improve clarity.

1. Figure 1 C. To make it easier for the reader it is suggested to move the legend beneath or next to the graphs.
2. Figure legend for Figure 2. Suggest to change description for D and E to "Hep-2 staining with (D) 1ug/ml of mAb (red) or (E) 1:200 diluted plasma (red)."
3. Figure 7B. The plus and minus are difficult to distinguish. Suggest to make them in larger font and also different colours, or alternatively colour-code the cells of the table.

Reviewer #2 (Comments to the Authors (Required)):

While severe COVID-19 is often associated with elevated autoantibody titers, the underlying mechanism behind their generation has remained unclear. This study identifies antibodies sharing genetic features with CoV1804 in COVID-19 patient-derived immunoglobulins, thereby constituting a novel public antibody, and potentially CL-reactive precursors may have developed multiple self-reactivities through convergent clonal selection and somatic mutation in response to viral antigens. While the paper provides an interesting perspective on the topic, I feel that there are some areas where the argument could be strengthened. For example, 1, should provide the data on gender-sex differences in autoantibodies testing. 2, If it possible, need to provide the data about proportion and quantity characteristics of plasmablasts in COVID-19 patients by flow cytometry analysis. 3, In line 485 of Page 18, atmosphere of 8% CO₂ in air, normally 5% CO₂, please make sure.

Reviewer #3 (Comments to the Authors (Required)):

This article sheds light on the clonal composition and diversity of autoantibodies in the humoral response of SARS-COV-2 thereby revealing the nature of autoantibody production during COVID-19 and providing novel insights into the origin of virus-induced antibodies.

- In result section 1, line 172 is stated "after treatment, anti-S titers in patients increased significantly, but somewhat less than those of vaccinated donors" after treatment should be clarified, which type of treatment is meant, also not in materials and methods or supplementary data.
- Result section 2, the rationale for cardiolipin antibodies is somewhat missing? Should be clarified in more detail, also the rationale for SLE.
- General comment: anti-S, anti-N, anti-ORF8 antibodies (abbreviations are not present in text, ORF-8 can also use some introduction.
- I do not fully agree with the statement " a public clonotype repertoire of autoantibodies have not been reported". For in Rheumatoid arthritis public clonotypes of anti-citrullinated protein antibodies (ACPAs) have been identified. This should be clarified in more detail.
- At line 338: ... we investigated the reactivity of the representative clones against structurally different substances, including dsDNA, insulin and LPS. Why in particular these 3 substances?
- In table S1, how are the conditions severe and moderate determined, is this according to WHO criteria?

Reviewer #1 (Comments to the Authors (Required)):

This manuscript by Sakakibara et al. comprehensively describes clonal composition and diversity of SARS-CoV-2 reactive and cardiolipin reactive autoantibodies detected in severely ill patients. This is a very well-written manuscript that brings forth several important findings that are supported by the data. A major finding is description of a novel public antibody that recognises both viral and self antigen. Collectively this study provides fundamentally important insight into infection-associated autoimmunity.

(Response) We are grateful for the comments from Reviewer #1.

Only minor edits are suggested to improve clarity.

1. Figure 1 C. To make it easier for the reader it is suggested to move the legend beneath or next to the graphs.

(Response) As suggested by Reviewer #1, we modified Figure 1C in the revised manuscript.

2. Figure legend for Figure 2. Suggest to change description for D and E to "Hep-2 staining with (D) 1ug/ml of mAb (red) or (E) 1:200 diluted plasma (red)."

(Response) We modified the figure legend as suggested (line 949, of the revised manuscript).

3. Figure 7B. The plus and minus are difficult to distinguish. Suggest to make them in larger font and also different colours, or alternatively colour-code the cells of the table.

(Response) We made them in larger font and colored differently in Figure 7B of the revised manuscript.

Reviewer #2 (Comments to the Authors (Required)):

While severe COVID-19 is often associated with elevated autoantibody titers, the underlying mechanism behind their generation has remained unclear. This study identifies antibodies sharing genetic features with CoV1804 in COVID-19 patient-derived immunoglobulins, thereby constituting a novel public antibody, and potentially CL-reactive precursors may have developed multiple self-reactivities through convergent clonal selection and somatic mutation in response to viral antigens. While the paper provides an interesting perspective on the topic, I feel that there are some areas where the argument could be strengthened. For example, 1, should provide the data on gender-sex differences in autoantibodies testing. 2, If it possible, need to provide the data about proportion and quantity characteristics of plasmablasts in COVID-19 patients by flow cytometry analysis. 3, In line 485 of Page 18, atmosphere of 8% CO₂ in air, normally 5% CO₂, please make sure.

(Response) We thank Reviewer #2 for his/her constructive comments.

1. We agree with this comment. Investigating biological sex differences in functional assessments enhances the rigor of research.

The below graph shows anti-CL autoantibody titers for males and females in each group in this study. In male donors, anti-CL titers of COVID-19 patients at ICU admission (COVID-19, T1) were significantly higher than those of healthy individuals before and after vaccination. On the other hand, in females, no significant difference in anti-CL titers was detected between COVID-19 patients and vaccinated donors. This might be related to the fact that males tend to suffer from more severe COVID-19 compared to females (Kharroubi and Diab-EI-Harake, 2022; Vahidy et al., 2021). We added this result as new Figure S4 to the revised manuscript and mentioned the sex difference in autoantibody production in line 201, in the main text of the revised manuscript.

2. Unfortunately, there are no available PBMC samples collected during the same periods that can be analyzed by flow cytometry. Previous studies have reported that plasmablasts are robustly increased in COVID-19 patients as noted in the original manuscript (line 109, of the revised manuscript).

The current study used the data from the Osaka scRNAseq cohort, in which plasmablasts, as defined by single-cell transcriptomics, accounted for 10% and 12% of B cells in moderate and severe patients, respectively (Edahiro et al., 2023). This proportion was consistent with a previous study of CyTOF-based analysis for patients' PBMCs obtained at the same hospital during the same period as our study. The study has reported that the proportion of plasmablasts was significantly higher than in controls and represented about 10% of B cells in the peripheral blood of severe COVID-19 patients (Søndergaard et al., 2023). These indicates a dramatical increase of plasmablasts in severe COVID-19 patients as these cells are typically around 1% of blood B cells in healthy individuals.

3. We are grateful for this feedback. In accordance with the manufacturer's

recommendation, Expi293F cells were maintained in an atmosphere of 8% CO₂; however, other cell lines used in this study were cultured in a 5% CO₂ environment. We revised the Method section of "Cell lines" as below (line 515, of the revised manuscript):

"Expi293F cells (Thermo) were cultured in Expi293 Expression Medium (Gibco) in an incubator with humidified atmosphere of 8% CO₂ in air on an orbital shaker rotating at 125 rpm at 37 °C. HEK293T (ATCC), Vero E6 TMPRSS2 (JCRB1819; NIBION) and HEp-2 (ATCC) were cultured in Dulbecco's modified Eagle medium (DMEM) supplemented with 10% FBS, 1 mM sodium pyruvate, 100 U/ml of penicillin-streptomycin in an incubator with humidified atmosphere of 5% CO₂ in air at 37°C."

References

Edahiro et al., Nat Gent 2023, doi: 10.1038/s41588-023-01375-1
Kharroubi, and Diab-El-Harake. Front Public Health 2022, doi: 10.3389/fpubh.2022.1029190
Søndergaard et al., Pro Natl Acad Sci U S A, 2023, doi: 10.1073/pnas.2217902120
Vahidy et al., Plos One 2021, doi: 10.1371/journal.pone.0245556

Reviewer #3 (Comments to the Authors (Required)):

This article sheds light on the clonal composition and diversity of autoantibodies in the humoral response of SARS-COV-2 thereby revealing the nature of autoantibody production during COVID-19 and providing novel insights into the origin of virus-induced antibodies.

(Response) We appreciate the insightful comments from Reviewer #3.

- In result section 1, line 172 is stated "after treatment, anti-S titers in patients increased significantly, but somewhat less than those of vaccinated donors" after treatment should be clarified, which type of treatment is meant, also not in materials and methods or supplementary data.

(Response) We are sorry for causing confusion on this sentence. At first, vaccinated donors did not have any treatment. We modified the sentence as below (line 180, of the revised manuscript):

"Anti-S titers in patients increased significantly after treatment, although these antibody titers were somewhat lower than those of healthy donors following vaccination (Figure S3B)."

We then added a brief description about the treatment for COVID-19 in Materials and Methods as below (line 465, of the revised manuscript):

"These patients received intensive care, including mechanical ventilation, anticoagulation therapy, and corticosteroids, in the ICU."

- Result section 2, the rationale for cardiolipin antibodies is somewhat missing? Should be clarified in more detail, also the rationale for SLE.

(Response) It has been shown that antibodies against phospholipids including CL are elevated in severe COVID-19 patients and play a role in coagulopathy (Zuo et al., 2020). We have mentioned this in Introduction of the original manuscript (line 78). However, based on this comment, we added a brief explanation of the rationale behind testing for CL antibodies and their relevance to SLE at the beginning of the second section of Results as below (line 191, of the revised manuscript):

"CL antibody tests generally aid in the evaluation of systemic lupus erythematosus (SLE) and assess the risk of complications; high anti-CL antibody titers are associated with increased risk of thrombosis (Tarr et al., Clin Rev Allergy Immunol, 2007). Several studies have shown that autoantibodies to CL and other phospholipids are elevated in severe COVID-19 patients, suggesting their potential contribution in the development of COVID-19-associated coagulopathy during the early pandemic. "

- General comment: anti-S, anti-N, anti-ORF8 antibodies (abbreviations are not present in text, ORF-8 can also use some introduction.

(Response) We appreciate this comment. We added these abbreviations at the points where the corresponding terms first appear in the main text (line 129).

- I do not fully agree with the statement " a public clonotype repertoire of autoantibodies have not been reported". For in Rheumatoid arthritis public clonotypes of anti-citrullinated protein antibodies (ACPAs) have been identified. This should be clarified in more detail.

(Response) Upon this comment, we revised the sentence as below (line 244):

"Although viral infections often induce convergent or public antibody responses, public clonotype repertoires of autoantibodies, defined as antibodies sharing genetic features and structural modes of antigen recognition across multiple individuals, are relatively uncommon."

- At line 338: ... we investigated the reactivity of the representative clones against structurally different substances, including dsDNA, insulin and LPS. Why in particular these 3 substances?

(Response) Some PA-N-CoV1804 antibodies were found to bind not only to SARS-CoV-2 N but also to CL and a nucleolar antigen, indicating their multiple reactivity or polyreactivity. Polyreactivity is defined by ability of antibodies to bind structurally different antigens. We selected dsDNA (nucleic acid), insulin (peptide) and LPS (liposaccharide) as test antigens

for polyreactivity based on previous studies in this field (Yurasov et al., 2005; Guthmiller et al., 2020).

The original text may lack the rationale behind performing the polyreactive assay. In light of this comment, we modified the beginning of the section in the revised manuscript (line 355).

- In table S1, how are the conditions severe ad moderate determined, is this according to WHO criteria?

(Response) We have determined the conditions of patients in accordance to WHO criteria (WHO Working Group on the Clinical Characterisation and Management of COVID-19 infection, 2020), which are now mentioned in the Materials and Methods section (line 463, of the revised manuscript).

References

- Tarr et al., Clin Rev Allergy Immunol., 2007, doi: 10.1007/s12016-007-0009-8
Zuo et al., Sci Trans Med 2020, doi: 10.1126/scitranslmed.abd3876.
Yurasov et al., J Exp Med 2005, doi: 10.1084/jem.20042251
Guthmiller et al., Immunity, 2020, doi: 10.1016/j.immuni.2020.10.005
WHO Working Group on the Clinical Characterisation and Management of COVID-19 infection, Lancet Infect Dis 2020, doi: 10.1016/S1473-3099(20)30483-7

September 4, 2024

RE: Life Science Alliance Manuscript #LSA-2024-02774R

Dr. Shuhei Sakakibara
Osaka University
Immunology Frontier Research Center
3-1 Yamada-Oka
Suita, Osaka 565-0875
Japan

Dear Dr. Sakakibara,

Thank you for submitting your revised manuscript entitled "Clonal landscape of autoantibody-secreting plasmablasts in COVID-19 patients". We would be happy to publish your paper in Life Science Alliance pending final revisions necessary to meet our formatting guidelines.

- please be sure that the authorship listing and order is correct
- for the figure 6 legend, we encourage you to introduce the figure panels in alphabetical order
- please add a callout for Figure 6E in your main manuscript text

A. FINAL FILES:

B. MANUSCRIPT ORGANIZATION AND FORMATTING:

Sincerely,

September 6, 2024

RE: Life Science Alliance Manuscript #LSA-2024-02774RR

Dr. Shuhei Sakakibara
Osaka University
Immunology Frontier Research Center
3-1 Yamada-Oka
Suita, Osaka 565-0875
Japan

Dear Dr. Sakakibara,

Thank you for submitting your Resource entitled "Clonal landscape of autoantibody-secreting plasmablasts in COVID-19 patients". It is a pleasure to let you know that your manuscript is now accepted for publication in Life Science Alliance. Congratulations on this interesting work.

DISTRIBUTION OF MATERIALS:

Again, congratulations on a very nice paper. I hope you found the review process to be constructive and are pleased with how the manuscript was handled editorially. We look forward to future exciting submissions from your lab.

Sincerely,
